# Multi-cohort, cross-species urinary proteomics reveals signatures of LRRK2 dysfunction in Parkinson's disease

Duc Tung Vu [1], William Sibran[2], Andreas Metousis [1], Laurine Vandewynckel[2], Basak Eraslan[3], Liesel Goveas [2], Ericka CM Itang [1], Claire Deldycke[2], Adriana Figueroa-Garcia [2], Réginald Lefèbvre[2], Johannes Bruno Müller-Reif [1], Sebastian Virreira Winter [4], Marie-Christine Chartier-Harlin [2✉], Jean-Marc Taymans [2✉], Matthias Mann [1✉] & Ozge Karayel [1,5✉]

## Abstract

Pathogenic mutations in Leucine-rich repeat kinase 2 (*LRRK2*) are the predominant genetic cause of Parkinson's disease (PD) and often increase kinase activity, making LRRK2 inhibitors promising treatment options. Although LRRK2 kinase inhibitors are advancing clinically, non-invasive readouts of LRRK2-linked pathway modulation remain limited. Profiling urinary proteomes from 1215 individuals across three cohorts and integrating whole-genome sequencing from >500 participants to map genotype–proteome associations, we identified 177 urinary proteins associated with pathogenic LRRK2, enriched for lysosomal/glycosphingolipid, immune, and membrane-trafficking pathways. Machine learning narrowed the features to a cohort-agnostic 30-protein panel that classified G2019S carriers with a mean ROC AUC of 0.91 across independent tests. To evaluate translation, we performed multi-organ and urinary proteomics in rat gain- and loss-of-function models (BAC-*LRRK2*[G2019S] and *Lrrk2*[KO]) and after Lrrk2 inhibition (MLi-2 and PF-475), revealing tissue-specific responses—strongest in kidney—and cross-species overlap, including 24 brain proteins detectable in human urine. Rat-derived perturbations predicted *LRRK2* mutation status in patients (AUC 0.75) and reversed with Lrrk2 inhibition, supporting their pharmacodynamic utility. Together, our findings establish urine as a scalable, non-invasive matrix that captures systemic and brain-relevant consequences of LRRK2 dysfunction and nominate candidate pharmacodynamic markers set to support LRRK2-directed trials.

**Keywords** LRRK2; Parkinson's Disease; Urine; Proteomics; Biomarker
**Subject Categories** Biomarkers; Neuroscience; Proteomics

## Introduction

Parkinson's disease (PD) is the second most common neurodegenerative disorder, affecting about 0.2% of the general population and 2% of those aged over 70 (Brakedal et al, 2022; Marras et al, 2018; Tysnes and Storstein, 2017). Clinically, PD is marked by progressive dopaminergic neurodegeneration, leading to motor and non-motor symptoms that are crucial yet challenging for diagnosis (Aerts et al, 2011; Jankovic, 2008). Although genetic, environmental, and biochemical factors contribute to disease risk, genome-wide association studies (GWAS) have identified numerous genetic risk loci. Among these, mutations in *LRRK2*, most prominently G2019S substitution, represent the most common cause of autosomal-dominant PD and are also found in sporadic cases, implicating shared molecular pathways across PD subtypes (Kluss et al, 2019).

Studies in the last decade have significantly advanced our understanding of how *LRRK2* mutations might contribute to PD pathogenesis (Alessi and Sammler, 2018; Li et al, 2014; Taymans, Fell, et al, 2023). *LRRK2* encodes a large multidomain protein with GTPase and kinase activities (Alessi and Pfeffer, 2024). The common pathogenic mutations are found in the enzymatic domains (Goveas et al, 2021; Taylor and Alessi, 2020), underscoring their role in the pathogenicity of LRRK2-driven PD. These mutations often lead to increased kinase activity, resulting in heightened phosphorylation of Rab GTPases (Atashrazm et al, 2019; Ö. Karayel et al, 2020; Steger et al, 2016, 2017). LRRK2 hyperactivation disrupts vesicular trafficking (Pan et al, 2017; Piccoli et al, 2011; Rivero-Ríos et al, 2019), autophagy (Bravo-San Pedro et al, 2013; Manzoni and Lewis, 2017), and lysosomal function (Bonet-Ponce and Cookson, 2022; Ysselstein et al, 2019), contributing to neuronal stress, inflammation, and degeneration (Cook et al, 2017; Wallings et al, 2020). These insights have catalyzed the development of LRRK2-targeted therapeutics that diminish LRRK2 activity (Atashrazm and Dzamko, 2016),

[1]Department of Proteomics and Signal Transduction Max Planck Institute of Biochemistry, Am Klopferspitz 18, 82152 Martinsried, Germany. [2]Univ. Lille, Inserm, CHU Lille, UMR-S 1172—LilNCog—Lille Neuroscience & Cognition, F-59000 Lille, France. [3]ML and Bioinformatics, Arc Institute, Palo Alto, CA CA94304, USA. [4]ions.bio GmbH, Am Klopferspitz 19, 82152 Planegg, Germany. [5]Present address: Genentech Inc, South San Francisco, CA 94080, USA. ✉E-mail: marie-christine.chartier-harlin@inserm.fr; jean-marc.taymans@inserm.fr; mmann@biochem.mpg.de; oezgekarayel@gmail.com

including small-molecule inhibitors (Fell et al, 2015) or antisense oligonucleotides (Taymans and Greggio, 2016; Zhao et al, 2017), several of which are now in late-stage clinical trials (Morez et al, 2024; Tolosa et al, 2020), increasing the need for scalable, non-invasive biomarkers that report on-target pathway modulation in vivo.

The current LRRK2 biomarker toolkit includes phosphorylation-based readouts (LRRK2 Ser935/Ser1292 and the substrate Rab10 Thr73 phosphorylations) and lipid measures such as urinary bis(monoacylglycerol)phosphates (BMPs; particularly di-22:6-BMP). These markers are valuable for confirming target engagement and optimizing dose, but they offer limited insight into the broader molecular consequences of LRRK2 dysregulation, therapeutic targeting or genotype-resolved biology (Alcalay et al, 2020; Gomes et al, 2023; Jennings et al, 2022, 2023).

Urine is a practical, non-invasive biofluid that can be sampled longitudinally at scale and has emerged as a promising matrix for discovery of protein biomarkers in neurodegenerative diseases (Hadisurya et al, 2023; Kumar et al, 2025; Rideout et al, 2020a; Rutledge et al, 2024; Virreira Winter et al, 2021). Using high-coverage MS-based proteomics, we and others have identified distinct urinary proteomic profiles in *LRRK2*[G2019S] carriers, including the upregulation of lysosomal and glycosphingolipid metabolism proteins. However, the full potential of urinary proteomic alterations to serve as systems-level readouts of PD risk variants beyond *LRRK2*[G2019S] and as pharmacodynamic biomarkers of on-target pathway modulation and treatment response remains underexplored and requires validation in larger populations.

Here, we performed an in-depth analysis of the urinary proteomes from 1215 individuals with PD and/or *LRRK2* mutations across three independent cohorts, alongside *Lrrk2* knockout (KO) rats, bacterial artificial chromosome (BAC)-*LRRK2*[G2019S] rats, and rats treated with the Lrrk2 inhibitors MLi-2 or PF-475. Integrating these cross-cohort and cross-species datasets with machine learning, we identified robust, cohort-agnostic urinary protein signatures of LRRK2 dysfunction that are sensitive to—and reversible with—kinase inhibition, supporting the development of LRRK2-directed therapies in PD. In parallel, we analyzed whole-genome sequencing data from >500 participants to map how 58 PD risk variants across 45 loci—beyond *LRRK2*[G2019S], which showed the strongest effect—shape the urinary proteome, revealing correlations in alterations across genotypes. Our study establishes urine as a scalable, non-invasive matrix for monitoring LRRK2 pathway modulation and target engagement at the proteome level, complementing lipid and phosphorylation biomarkers.

# Results

## Investigating urinary proteome alterations associated with genetic variants in 45 PD genes

To investigate how PD and its most common autosomal-dominant genetic cause, the *LRRK2*[G2019S] mutation, affect the urinary proteome, we previously conducted MS-based profiling in two small urine cohorts (Virreira Winter et al, 2021): (i) the Columbia cohort with 120 individuals, including 44 *LRRK2*[G2019S] carriers, and (ii) the LRRK2 Cohort Consortium (LCC) with 115 individuals, of

which 55 were *LRRK2*[G2019S] carriers (Fig. 1A). Our studies pinpointed unique protein signatures linked to lysosomal and glycosphingolipid metabolism dysfunction that could serve as biomarkers for stratifying *LRRK2* mutation carriers.

To validate these findings in a larger population and assess the extent of proteomic changes associated with genetic PD, we here analyzed in-depth the largest urinary proteomics study to date, involving participants from the observational, multicenter clinical study known as the Parkinson's Progression Markers Initiative (PPMI) (Marek et al, 2018). The baseline study includes 980 individuals—healthy controls, PD patients, and prodromal PD patients—of whom 305 were *LRRK2*[G2019S] carriers (Fig. 1A). To prepare urinary proteomes for MS analysis, proteins were extracted from neat urine samples and digested into peptides using a modified MStern blotting protocol (Berger et al, 2015). To achieve high proteome coverage, peptides were separated using a 30-sample-per-day (SPD) Evosep gradient. Data were collected on a Bruker timsTOF Pro mass spectrometer in diaPASEF mode (Meier et al, 2021) and analyzed using DIA-NN (Demichev et al, 2020), detecting a total of 6479 proteins with a remarkable median depth of 3765 proteins per individual (Appendix Fig. S1A).

To assess reproducibility across the data acquisition process, pooled reference samples were injected in every 12th position on each plate alongside the study samples. Coefficients of variation (CVs) were calculated within each plate, showing high reproducibility for the pooled samples, with a median CV of 25% across plates (Appendix Fig. S1B). In contrast, the mean biological variance across patient samples was 68%, allowing for straightforward extraction of biological pathway alterations from the technical variation (Appendix Fig. S1C). Principal component analysis (PCA) revealed tight clustering of the pooled reference samples, further indicating high technical consistency and overall data quality (Appendix Fig. S1D). Across multiple plates, UMAP visualization showed no evidence of plate-associated heterogeneity, indicating minimal batch effects (Appendix Fig. S1E). To maintain high data quality, we evaluated each sample in the three studies for blood contamination and technical variation, leading to the exclusion of 54 samples in total: seven from the Columbia cohort, four from the LCC cohort, and 43 from the PPMI cohort (Appendix Fig. S1F,G). UMAP analysis following quality control measures identified sex as the primary driver of variance in the urinary proteome (Appendix Fig. S1H).

The PPMI cohort includes high-quality whole-genome sequencing (WGS) of blood-derived DNA accessible from 534 out of 980 individuals who contributed urine samples for proteomic analysis. This enabled us to investigate the impact of not only the *LRRK2*[G2019S] mutation but also 57 additional variants across 45 PD-associated loci on the urinary proteome. To isolate the contribution of each genetic variant while accounting for confounding factors such as age and sex that influence protein abundance, we performed multiple linear regression analysis for each gene across the cohort (Methods and Dataset EV1). The aggregated analysis revealed that genetic variants in *LRRK2* had the most pronounced impact on the urinary proteome, with nearly 300 proteins showing significant differential regulation (Fig. 1B). Interestingly, variants within the *ZNF646/KAT8/BCKDK* and *MIR4697* regions produced proteomic signatures similar to *LRRK2*[G2019S], while mutations in *SATB1*, *CTSB*, and *GBA* were significantly associated with opposite profiles (Fig. 1C).

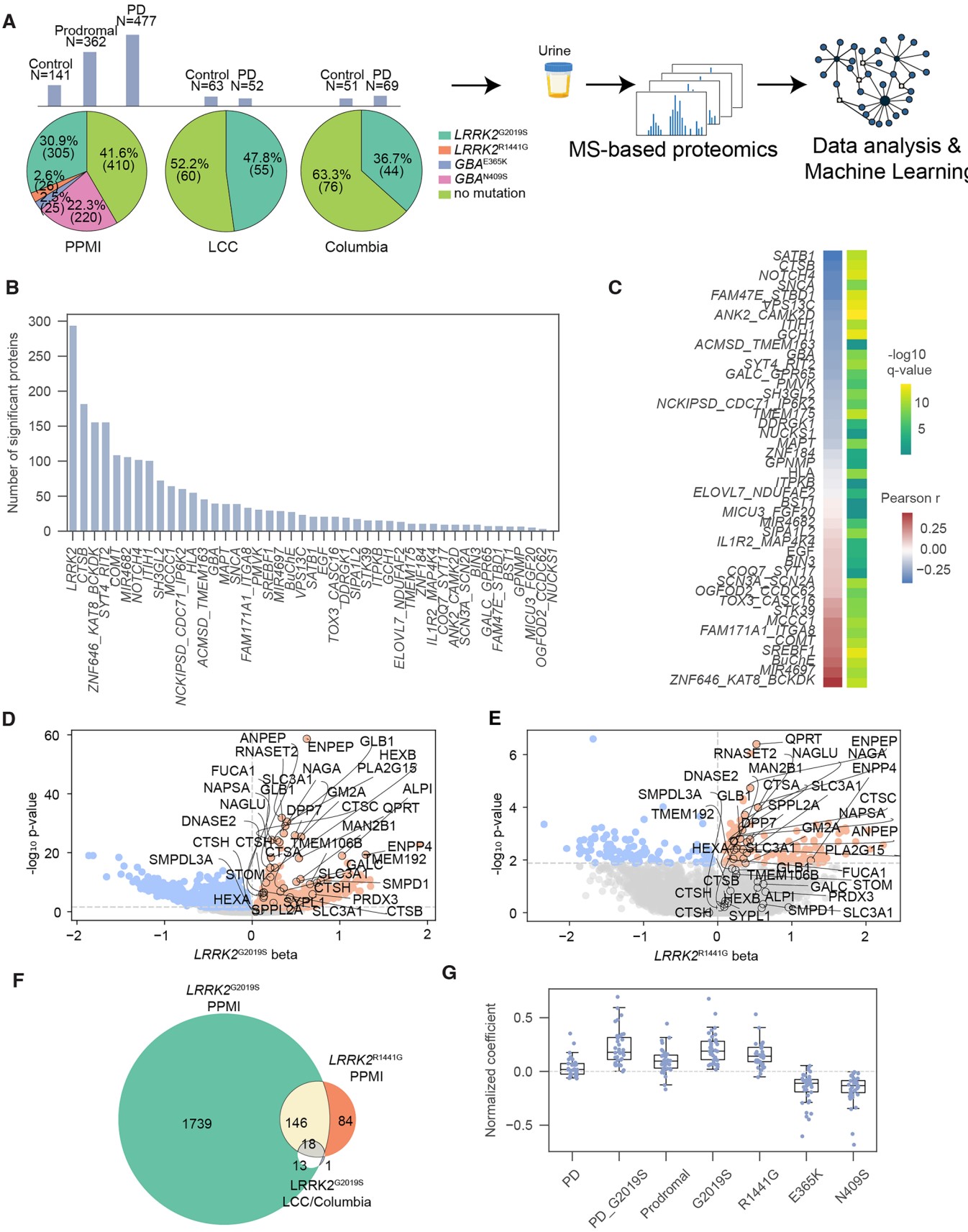

**Figure 1.   Validating pathogenic LRRK2-related proteome alterations in human urine.**

(A) Overview of patient cohorts where urine samples tested in this study originate from. Three sample collections were used: the Parkinson's progression markers initiative (PPMI), LRRK2 cohort consortium (LCC) and Columbia. In the top part of the panel, histograms indicate number of samples from control, prodromal (for PPMI) and Parkinson's subjects for each sample collection. Pie charts in the lower part of the panel indicate for each sample collection the proportions of subjects with *LRRK2* mutations, *GBA* mutations or no mutations (see color-coded legend, bottom right). (B) Distribution of significantly and differentially abundant proteins (q value <5%) identified across different genetic variants from WGS data compared to WT controls using a multiple linear regression approach. The WGS data were grouped and aggregated by the sum for mutations affecting the same gene locus and used as a confounding variable in the regression analysis. *P*- and *q*-values were extracted for each gene's beta value. (C) Pearson correlation coefficient and corresponding $-\log_{10} q$ values of beta-value profile derived from multiple linear regression analysis between *LRRK2* and other aggregated genetic risk variants. (D, E) Volcano plots of $-\log_{10} p$ values vs. beta values from multiple linear regression for G2019S (D) and R1441G (E) with $n = 6479$ proteins. The horizontal dotted line marks the 5% q value cutoff. The vertical dotted line separates the proteins into up- and downregulation. The significantly regulated proteins in LCC and Columbia are annotated. (F) Venn diagram showing the overlap of pathogenic LRRK2-regulated proteins (177 in total with at least one intersection) (q value <5%) across cohorts (LCC, Columbia and PPMI). (G) Distribution of normalized beta values from PPMI for the significant proteins in both LCC and Columbia ($n = 32$) across different genetic variants of *LRRK2* (G2019S and R1441G), *GBA* (N409S and E365K) and disease states (PD, PD with G2019S, and Prodromal). Boxplots display the median (center line), interquartile range (IQR; box bounds representing the 25th and 75th percentiles), and whiskers extending to 1.5× IQR from the box edges. Data points beyond the whiskers are shown as individual outliers. Beta values were normalized by division with the maximum beta value within each confounding variable.

Among the variants, the *LRRK2*[G2019S] mutation exhibited the strongest effect, followed by *GBA*[N409S] and *LRRK2*[R1441G] (Fig. EV1A). However, the WGS-based analysis was limited to only 534 individuals. To ensure comprehensive coverage of *LRRK2* and *GBA* mutations across all participants, we conducted a complementary analysis using the genetics consensus data, which provides complete *LRRK2* and *GBA* mutational status for all 980 patients (Dataset EV2). By applying our multiple linear regression approach to all individuals, we identified 1916 significantly changing proteins for *LRRK2*[G2019S] at a 5% *q* value (Fig. 1D) and 248 significant proteins for *LRRK2*[R1441G] (Fig. 1E). Interestingly, R1441G and G2019S displayed high correlation (Pearson r: 0.44 and q value: 0.0) indicating similar molecular mechanisms despite different variant types (Fig. EV1B). Enrichment analysis of G2019S- and R1441G-associated proteins in PPMI indicated compromised glycosphingolipid metabolism and immune system function (Fig. EV1C,D). Interestingly, similar to R1441G, three *COMT* variants showed proteomic signatures that closely resembled those of the *LRRK2*[G2019S] mutation (Fig. EV1B; rs165599: $r = 0.59$, $q = 0.0$; rs4633: $r = 0.47$, $q = 0.0$; rs165656: $r = 0.45$, $q = 0.0$). By contrast, *GBA*[E365K] and *GBA*[N409S] displayed inverse proteomic signatures (Fig. EV1B).

Notably, the urinary protein signature associated with *LRRK2*[G2019S] showed strong consistency across cohorts. Of the 32 significant proteins in both smaller cohorts (LCC & Columbia), nearly all (31 out of 32) ranked among the most differentially abundant proteins in the larger PPMI cohort (Fig. 1F). Providing positive controls, these included genes linked to PD and lysosomal biology, such as *GALC* (Senkevich et al, 2023), *PLA2G15* (Liu et al, 2024), *SMPD*1 (Alcalay et al, 2019), *TMEM106B* (Tropea et al, 2019), and *HEXA* and *HEXB* (Brekk et al, 2020). Interestingly, the abundance of these LRRK2[G2019S]-regulated proteins varied only with *LRRK2* mutation status, not with PD diagnosis itself (Figs. 1G and EV1E). Consistent with our previous analysis, *GBA* mutations—specifically E365K and N409S, present in 244 individuals in PPMI—affected the abundance of nearly all those proteins, but in the opposite direction (Figs. 1G and EV1E), underscoring the power of urinary proteomics in capturing genotype-specific disease mechanisms.

Finally, we examined the correlations between disease severity scores and the abundances of LRRK2[G2019S]-regulated urinary proteins in PPMI, Columbia and LCC. Proteins like ANPEP,

MAN2B1, FUCA1, ENPEP, PRDX3, and QPRT exhibited strong correlations with severity scores, including the Schwab and England Score and Hoehn and Yahr Scale, in PD patients with *LRRK2* mutations, but not PD lacking *LRRK2*[G2019S] (Fig. EV2). These proteins may serve as indicators of disease severity, facilitating the stratification of familial patients.

## Expanding the urinary proteomic signature of pathogenic *LRRK2* variants

Proteomic analysis of the PPMI cohort, comprising nearly 1000 individuals, validated our findings in smaller cohorts of *LRRK2*[G2019S] carriers. The large sample size substantially increased statistical power, allowing the identification of 146 additional proteins that were significantly altered in both G2019S and R1441G carriers. This expanded the total number of potential urinary protein markers associated with pathogenic *LRRK2* mutations to 177 (Fig. 1F; Dataset EV3). Of the 146 proteins uniquely identified in PPMI, 121 and 110 were also detected in the LCC or Columbia cohorts, respectively, although they did not reach statistical significance in both (Dataset EV3). For example, LAMP1, a lysosomal marker, was significantly altered in *LRRK2*[G2019S] carriers in LCC, but not in Columbia, while being notably upregulated in PPMI mutation carriers (Dataset EV3).

To explore relationships among the 177 regulated proteins, we constructed a protein network using a ridge regression (Fig. EV3A; Dataset EV2). Louvain clustering identified four main protein clusters (Fig. EV3B). Cluster 1 encompassed most of the proteins regulated by LRRK2[G2019S] across all three cohorts and, as expected, showed significant enrichment for lysosomal function and glycosphingolipid metabolism (Fig. EV3B,C). Consistent with studies linking LRRK2 to endolysosomal trafficking (reviewed in Bonet-Ponce and Cookson, 2022; Roosen and Cookson, 2016), immune responses (Hakimi et al, 2011; Wallings and Tansey, 2019), and glycolysis (Oun et al, 2023; Tang, 2020), Cluster 2 included proteins linked to immune responses and antigen presentation, along with the LRRK2 substrate RAB10. Cluster 3 was associated with gluconeogenesis and fructose metabolism, while Cluster 4 contained proteins related to endolysosomal trafficking and Golgi vesicles (Fig. EV3C). Ridge regression combined with the elbow method revealed that ten proteins captured ~40% of the network's variance ($R^2 = 0.396$, Appendix Fig. S2A–C; Dataset EV2). The top

ten proteins with the highest individual $R^2$ values spanned all four clusters: DPP7, NAGLU, and PLA2G15 in Cluster 1; IGHV3-11 in Cluster 2; ANPEP, ACY1, and ALDH1A1 in Cluster 3; and PTPRJ, SPINK1, and SPP1 in Cluster 4 (Fig. EV3B). This suggests the network is organized into functional groups of proteins with redundant contributions to the overall variance.

Together, we identified a reliable panel of *LRRK2*^[G2019S]-regulated urinary proteins functionally linked to immune processes, membrane trafficking, lysosomal function, and glycosphingolipid metabolism, exhibiting consistent regulation patterns across cohorts, highlighting their potential as LRRK2 biomarkers.

## A machine learning tool for accurately classifying *LRRK2* mutation status

Motivated by the substantial differences in urinary proteomes between *LRRK2* mutation carriers and wild-type (WT) allele carriers, we evaluated machine learning (ML) models to classify *LRRK2*^[G2019S] mutation status based on urinary proteome profiles. To enhance robustness, we combined samples from the three cohorts (PPMI, LCC, and Columbia), resulting in 7663 quantified proteins, split into a training set and cohort-specific test sets (Appendix Fig. S3). F-test and mutual information reduced the data to 128 key features (Dataset EV4), later sorted by importance using the coefficients from a support vector machine (SVM) classifier. We then incrementally added them and monitored model performance through a repeated stratified k-fold with 5 splits and 3 repeats cross-validation (Appendix Figs. S3 and S4). The top 30 discriminating features achieved excellent performance across all test cohorts: ROC AUC of 0.89 for PPMI, 0.89 for LCC, and 0.96 for Columbia (Fig. 2A), with average precision of 0.74 for PPMI, 0.92 for LCC, and 0.95 for Columbia (Fig. 2B). Our model correctly classified 80 out of 91 *LRRK2*^[G2019S] carriers and 137 out of 171 WT allele carriers, achieving a sensitivity of 88% and a specificity of 80% across all test datasets (Fig. 2C). These features included several lysosomal proteins, such as ENPEP, GM2A, NEU1, HEXB, CTSH, CTSB, PLD3, TMEM192, SCPEP1, and SMPD1 (Appendix Fig. S5). Notably, most of these proteins had already been identified as LRRK2-regulated through linear regression analysis in the PPMI cohort. This overlap indicates that the model-derived features are not arbitrary, but instead reflect biologically meaningful proteomic alterations associated with pathogenic LRRK2.

Additionally, we trained the model on the PPMI cohort alone and tested it on the LCC and Columbia cohorts, identifying similar key features—for Columbia: ENPEP, ALPI, NEU1, THBD, RNASET2, NGFR, DGCR2, TXN2, ANPEP, CTS, and for LCC: ENPEP, ALPI, NEU1, NGFR, and DGCR2 (Dataset EV4). The model achieved ROC AUCs of 0.83 for both cohorts (Appendix Fig. S6), validating these proteins as cohort-agnostic urinary biomarker candidates for predicting *LRRK2* mutation status.

Despite the high classification accuracy, the model misclassified 30 individuals in the PPMI test set as *LRRK2*^[G2019S] carriers (Fig. 2C). To explore the reason for these false positives, we applied UMAP dimensionality reduction to the 30 selected protein features. The model showed strong predictive certainty for these individuals, as reflected by the high predicted logits (Appendix Fig. S7A,B). Further investigation of mutation and disease states revealed 19 PD and eight prodromal patients. Remarkably, genetic profiling revealed that 20 carried alternative *LRRK2* variants (e.g., R1441G)

and 8 harbored *GBA* mutations (e.g., N409S, T408M). Interestingly, one false positive individual carried *LRRK2*^[N2081D] (Appendix Fig. S7C), a variant linked to increased inflammatory (IBD) risk and elevated kinase activity (Heaton et al, 2025). In addition, a single PD patient carried multiple genetic risk variants, including *COMT* variants, but no *LRRK2* or *GBA* mutation (Appendix Fig. S7E). The remaining cases lacked known *LRRK2* or *GBA* mutations (Appendix Fig. S7C,D). Although PPMI annotations may not comprehensively capture all *LRRK2* variants associated with PD, these findings suggest that proteomic and phenotypic similarities may extend beyond familial PD to idiopathic PD (Kalogeropulou et al, 2022) and, in specific cases, possibly to non-neurological conditions such as IBD.

The same ML pipeline was used to differentiate between PD patients and controls, as well as genetic PD and non-manifesting carriers. However, it performed poorly, only slightly better than random chance (Appendix Fig. S8A,B; Dataset EV4). We also assessed our model's ability to distinguish G2019S PD from non-G2019S PD. Remarkably, among the 30 features, 9 (MGAM, EPHB1, SLC35F6, NEU1, TMEM106A, ENPEP, PF4, ALPI, and CDC42) achieved ROC AUC of 0.84 for PPMI, 0.96 for LCC, and 0.89 for Columbia (Appendix Fig. S9; Dataset EV4), further confirming the model's robustness in classifying *LRRK2* mutation status.

Given the differences in instrumentation, sample collection and preparation, and data acquisition protocols across cohorts, we considered the possibility that batch effects could influence the outcome of our ML analysis combining the three cohorts (Appendix Fig. S10A; Dataset EV5). To address this, we applied the same pipeline to a batch-corrected dataset alongside the original and compared the outcomes. Feature selection yielded consistent results, with the top discriminative proteins–such as NEU1, ALPI, CTSB, ENPEP, and CTSH - shared between corrected and uncorrected datasets (Appendix Fig. S10B; Dataset EV4). Importantly, nearly all batch-corrected features were also identified as differentially abundant in our multiple linear regression analysis of *LRRK2*^[G2019S] carriers in the PPMI cohort, with half represented in the pathogenic LRRK2-regulated protein network. Model performance remained the same after batch correction (Appendix Fig. S10C), underscoring the robustness of our results and confirming that batch effects did not compromise feature selection or predictive accuracy.

## Pathogenic LRRK2-associated urinary proteins as candidate pharmacodynamic markers

We next leveraged rat preclinical models to determine whether urinary proteins regulated by pathogenic LRRK2 in humans are sensitive to Lrrk2 loss or pharmacological inhibition. We began by assessing whether comparable proteomic changes occur in urine from transgenic BAC-*LRRK2*^[G2019S] rats (Sloan et al, 2016). Using library-free DIA proteomics on urine from BAC-*LRRK2*^[G2019S] rats ($n = 8$) and controls ($n = 10$), we reproducibly identified 2486 proteins (Appendix Fig. S11A), with ~11% significantly altered at $q$ values <5% (Fig. 3A, Appendix Fig. S11A,B; Dataset EV6). Of the 177 human pathogenic LRRK2-regulated urinary proteins, 101 homologs were detected in rat urine, 38 of which were also significantly regulated in G2019S rats (Fig. 3A). Notably, several of these proteins, including Enpep, Gm2a, Neu1, Smpd1, Ctsh, Ctsb,

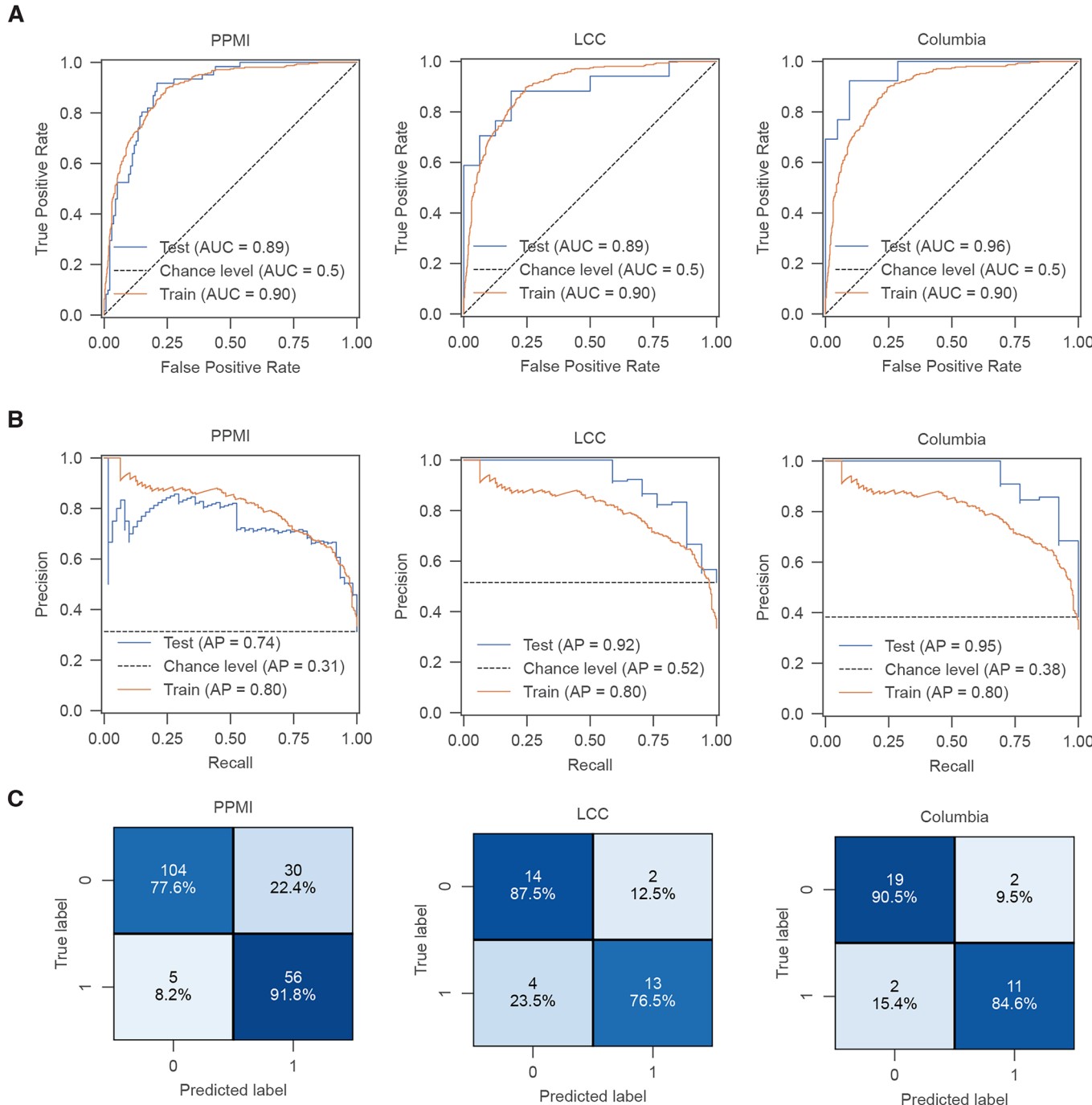

**Figure 2. Machine learning-based classification of LRRK2 mutation status.**

(A) Receiver operating characteristic (ROC) curves for a support vector machine (SVM) classifying individuals with *LRRK2*[G2019S] vs. controls using $n = 30$ proteins. The model was trained on combined urine data and tested on cohort-specific datasets: PPMI (left), LCC (middle), and Columbia (right). Random performance is shown by the dotted diagonal line. (B) Same as (A), but model performance was evaluated using a precision-recall curve. Average precision (AP) is annotated for the train and test datasets. (C) Confusion matrix for predictions in (A, B) depicting the number of true positive predictions in the lower right, true negative predictions in the upper left, false positive predictions in the upper right and false negative predictions in the lower left.

and Hexb, overlapped with ML-selected discriminators in humans, supporting conservation of the LRRK2-associated proteomics signature. Additionally, the altered proteins included Rab3a, a small GTPase involved in synaptic vesicle trafficking and a known LRRK2 substrate, and Smpd1, consistent with lysosomal/glyco-sphingolipid involvement. Regulated Neu1 and Ctsh further pointed to autophagy-lysosomal pathway perturbation characteristic of LRRK2-driven pathology.

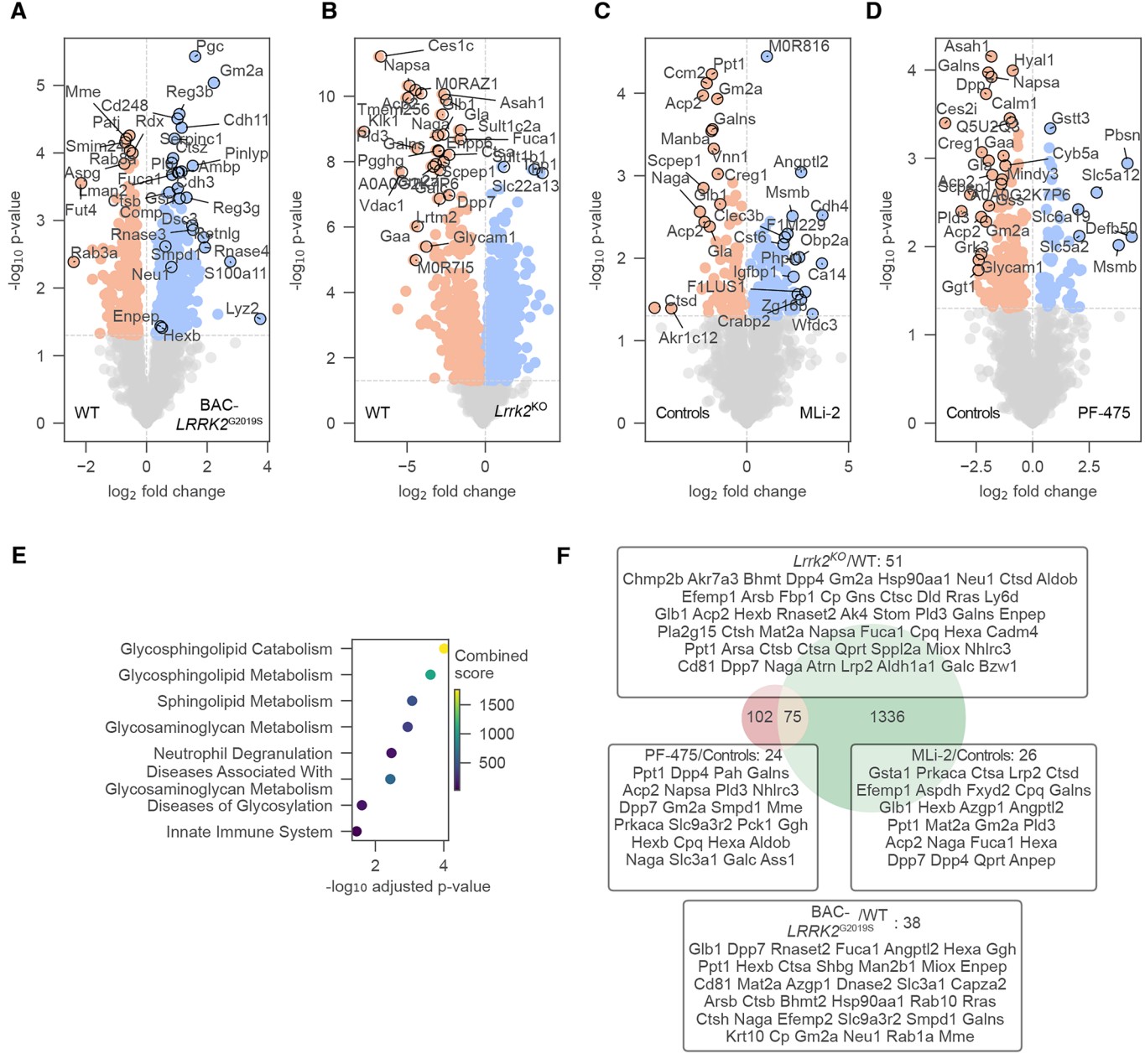

**Figure 3. Detecting proteome alterations associated with Lrrk2 loss, hyperactivation, and inhibition in rat urine.**

(A–D) Volcano plots comparing the urine proteomes of BAC-LRRK2[G2019S] vs. Lrrk2[WT] rats ($n = 1714$) (A) and Lrrk2[KO] vs. Lrrk2[WT] rats ($n = 1860$) (B), MLi-2 vs. Controls ($n = 2131$) (C), and PF-475 vs. Controls ($n = 2131$) (D) with dotted lines indicating 5% p value cutoffs. The top 30 most significant proteins with the highest fold-change are annotated. (E) Reactome enrichment analysis. Enrichment was performed on significant proteins (p value <5%) in at least two out of four comparisons. The combined score represents an aggregated metric calculated during enrichment analysis. (F) Venn diagram of union of significant proteins (p value <5%) in Lrrk2[KO] vs. Lrrk2[WT] or BAC-LRRK2[G2019S] vs. Lrrk2[WT] or MLi-2 vs. Controls or PF-475 vs. Controls (right) and network genes from human study (left). Intersecting proteins are annotated in a box with their respective comparison.

To determine sensitivity to genetic loss or pharmacological inhibition, we profiled urine from Lrrk2[KO] rats ($n = 9$), inhibitor-treated rats ($n = 7$; MLi-2 or PF-475) and littermate controls ($n = 13$). Both are Type I small-molecule inhibitors that target the ATP-binding site of LRRK2, stabilizing its active conformation (Fell et al, 2015; Henderson et al, 2015; Scott et al, 2017). They induce the dephosphorylation of the S935 site on LRRK2 and

suppress the phosphorylation of RAB proteins. MLi-2, an indoleazole derivative, exhibits nanomolar potency and high selectivity, while PF-475 features a pyrrolopyrimidine scaffold with slightly lower potency (Dang et al, 2025; Morez et al, 2024). Both compounds have shown robust inhibition of LRRK2 activity in vitro and in vivo and are widely used in preclinical research to study LRRK2-related disease mechanisms. Western blotting of

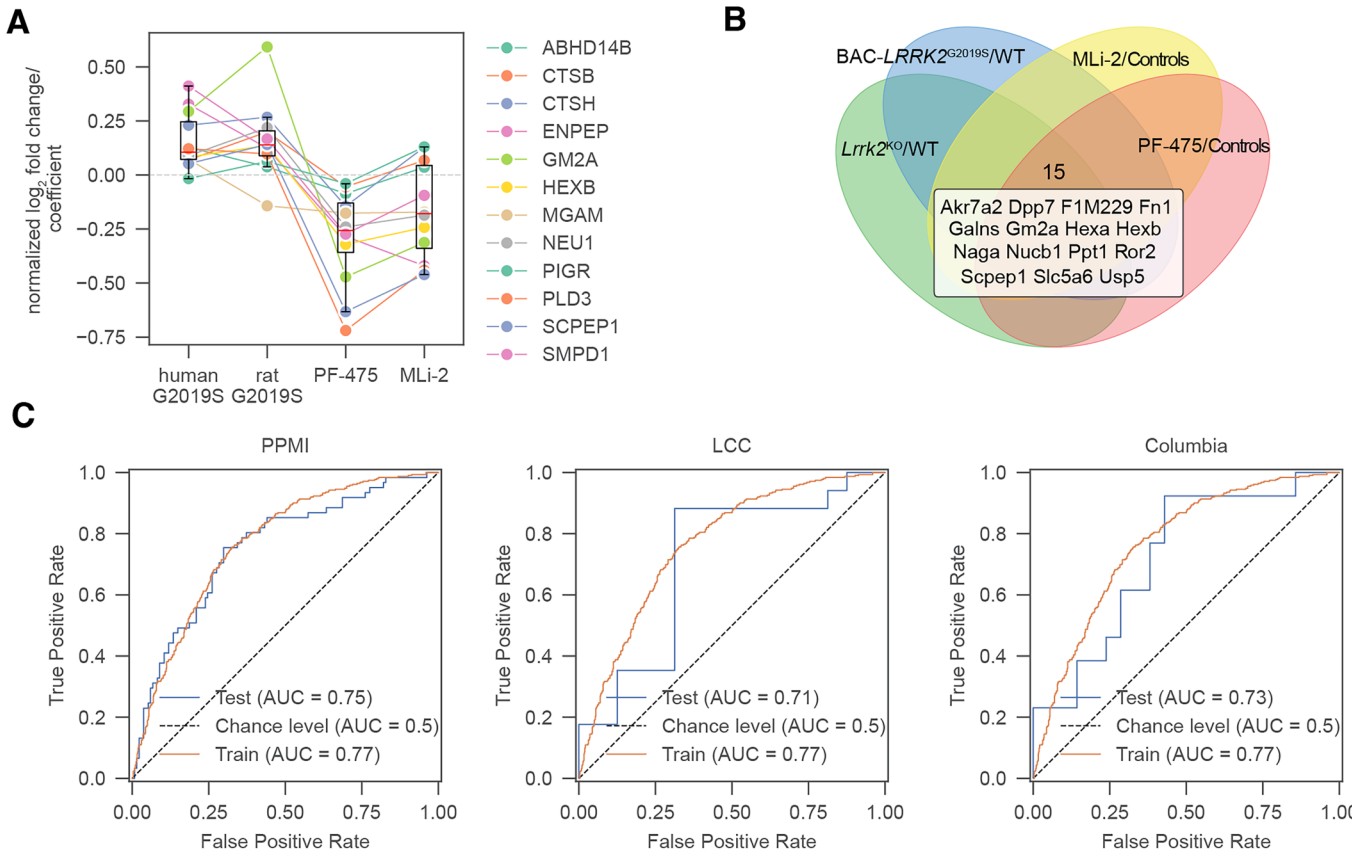

**Figure 4. Concordant regulation of LRRK2 biomarker from human urine study in rat urine with Lrrk2 modulation.**

(A) Analysis of biomarkers from the human study in rats ($n = 12$). The $\log_2$ fold-change of rat proteins and the regression coefficient of human proteins were normalized by the maximum $\log_2$ fold-change or regression coefficient to compare the regulatory directionality (up- or down-regulation). Boxplots display the median (center line), interquartile range (IQR; box bounds representing the 25th and 75th percentiles), and whiskers extending to 1.5× IQR from the box edges. Data points beyond the whiskers are shown as individual outliers. (B) Venn diagram of significant proteins ($p$ value <5%) between $Lrrk2^{KO}$ vs. $Lrrk2^{WT}$ or BAC-$LRRK2^{G2019S}$ vs. $Lrrk2^{WT}$ or MLi-2 vs. Controls or PF-475 vs. Controls. Proteins significant in all comparisons (15 in total) are annotated below the Venn diagram. (C) Receiver operating characteristic (ROC) curves for a support vector machine (SVM) classifying individuals with $LRRK2^{G2019S}$ vs. controls using the $n = 15$ intersecting proteins from the rat urine study. The model was trained on combined urine data and evaluated on cohort-specific test datasets: PPMI (left), LCC (middle), and Columbia (right). Random performance is shown by the dotted diagonal line.

kidney and brain tissues confirmed successful *Lrrk2* deletion and pharmacological inhibition (Appendix Fig. S12A,B).

Our MS-based proteomic analysis reproducibly identified 2635 proteins in the KO dataset and 2852 in the inhibitor dataset (Appendix Fig. S11A). In line with LRRK's diverse roles, $Lrrk2^{KO}$ exhibited the most significant differences compared to controls, with >50% of proteins significantly regulated (Fig. 3B; Appendix Fig. S11B; Dataset EV6). Fold-changes in KO vs. WT were strongly inverse to G2019S vs. WT (Pearson $r = -0.72$; Appendix Fig. S13A). MLi-2 and PF-475 treatment led to a consistent decrease in lysosomal proteins (Gla, Gaa, Gm2a, and Naga) (Fig. 3C,D). Regression analysis revealed changes induced by PF-475 and MLi-2 were highly concordant (Pearson $r = 0.71$; Appendix Fig. S13A), consistent with on-target Lrrk2 kinase inhibition. Inhibitor-induced urinary proteomics alterations resembled KO (Pearson $r = 0.50$) and anticorrelated with G2019S (Pearson $r = -0.20$) (Appendix Fig. S13A). Pathway analysis implicated lysosomes, glycosphingolipid metabolism, and immune programs (Fig. 3E), consistent with our findings in human studies. Importantly, 75 rat

urinary proteins regulated in at least one *Lrrk2* perturbation overlapped with the human signature, including Cdc42, Ctsb, Ctsh, Enpep, Gm2a, Hexb, Neu1, Pigr, Pld3, Scep1, Smpd1, and Sppl2a (Fig. 3F).

We further evaluated whether the ML–derived top 30 discriminative urinary proteins identified in humans are sensitive to Lrrk2 perturbation in rats. Of these 30 proteins, 12 were quantifiable in rat urine; largely all 12 were significantly upregulated in both human and rat G2019S and were down-regulated following Lrrk2 kinase inhibition with MLi-2 or PF-475 in rats (Fig. 4A). This inverse relationship between mutation-driven activation and pharmacological inhibition supports these proteins as candidate pharmacodynamic readouts of LRRK2-linked pathway modulation. Next, we identified a high-confidence signature of 15 urinary proteins that were consistently regulated across all Lrrk2 perturbations in rats (Fig. 4B). We used this consensus signature to train a support vector machine on the combined PPMI–LCC–Columbia human urinary proteomics dataset (Dataset EV4). Feature selection was deliberately restricted to these rat-

derived features to prevent data leakage and ensure an unbiased evaluation of the model's true predictive capacity across species. The classifier achieved ROC AUCs of 0.71–0.75 on independent cohort-specific test sets (Fig. 4C). Lastly, we evaluated model performance solely on proteins that are significantly regulated in G2019S rats, which resulted in 603 features (Dataset EV4). On these, the model achieved even better performance with ROC AUCs of up to 0.9 (Appendix Fig. S14). Thus, rat-derived Lrrk2-associated urinary signatures generalize to humans and may serve as a non-invasive indicator of pathway dysfunction and target engagement in clinical studies.

## Tissue-specific proteomic responses to Lrrk2 perturbation in rats

We next investigated how Lrrk2 perturbations—pathogenic hyperactivation (via expression of BAC-LRRK2$^{G2019S}$) and functional suppression (via genetic knockout or pharmacological inhibition)—remodel peripheral tissue proteomes in rats. To this end, we profiled the kidney, lung, and brain from the same cohorts of BAC-LRRK2$^{G2019S}$ and Lrrk2$^{KO}$ rats, as well as rats treated with the Lrrk2 inhibitors MLi-2 and PF-475. Across tissues, we identified over 9000 proteins (Appendix Fig. S12C). BAC-LRRK2$^{G2019S}$ and inhibitor-treated rats exhibited changes that were more modest than in knockouts yet consistent (Appendix Fig. S12D). These findings align with prior in vivo studies in Lrrk2$^{KO}$ mice and rats and in nonhuman primates treated with Lrrk2 inhibitors (Andersen et al, 2018; Baptista et al, 2013, 2020; Fuji et al, 2015; Kluss et al, 2021).

### Kidney

Lrrk2$^{KO}$ produced broad proteomic changes, including strong upregulation of lysosomal hydrolases (e.g., Galc, Naga, Hexa/Hexb, and Neu1) and downregulation of oxidative-stress enzymes (Akr1b10, Akr1b7, and Akr1b8) (Fig. 5A; Dataset EV6). BAC-LRRK2$^{G2019S}$ altered iron-transport and cytoskeletal proteins (Slc11a2, Septin4) (Fig. 5B). MLi-2 and PF-475 increased mitochondrial and exocytosis proteins (Fam210a and Scrn1) and modulated cytoskeletal organizers and vesicular-trafficking proteins (Ank3 and Arl3) (Fig. 5C,D). Multiple regulated proteins (Glb1, Wdr81, Sfxn2, Itsn2, Snx9, Lamp1, Cycs, Cpt1b, Sfxn3, and Hgs) matched prior reports by Kluss et al, 2021 (Fig. 5A–D). Fold-changes were negatively correlated between KO vs. WT and G2019S vs. WT, indicating that gain- and loss-of-function perturbations induce opposing molecular phenotypes (Appendix Fig. S13B). Both inhibitors elicited consistent alterations (Appendix Fig. S13B), with enrichment for lysosomal and mitochondrial pathways (Fig. 5E) and substantial overlap with G2019S-associated urinary protein changes in humans (Fig. 5F).

### Lung

Lrrk2$^{KO}$ regulated stress-response (e.g., Akr1b8), immune (Cd200, Anpep), and lysosomal (e.g., Gaa) proteins (Fig. EV4A), mirroring features seen in the kidney. G2019S increased neuroprotective/detoxification components (Lifr, Gstm1) and Lrrk2 itself (Fig. EV4B; Dataset EV6). MLi-2 modulated trafficking and apoptosis proteins (Tom1l1, Ppp1r13b, and Numbl) and the mitochondrial porin Vdac2, consistent with Kluss et al, 2021 (Fig. EV4C). PF-475 regulated calcium-homeostasis and respiratory-chain components

(e.g., Ndufa5 and Atp2b2) (Fig. EV4D). Both inhibitors produced robust but partially overlapping changes (Appendix Fig. S13C). Pathway enrichment highlighted cilia and secretory-granule biology (Fig. EV4E). Cross-species overlap with human urinary data was smaller than in the kidney but included Anpep, Enpep, Fam174a, Aldop, and Gsta1 (Fig. EV4F).

### Brain

LRRK2 dysfunction elicited neuron-specific responses distinct from peripheral tissues. Lrrk2$^{KO}$ upregulated Nefm and downregulated inflammatory mediators (S100a8/a9) and the mitochondrial complex I subunit Mt-nd2 (Fig. EV5A; Dataset EV6). BAC-LRRK2$^{G2019S}$ increased synaptic/signaling proteins, including the neuropeptide Rln3 (Fig. EV5B). As in the lung, the two inhibitors produced partially overlapping profiles: MLi-2 enhanced membrane-associated proteins (Palmd, Tmem245) (Fig. EV5C), whereas PF-475 altered mitochondrial energy-metabolism proteins (Ckmt1 and Chdh), the ubiquitin-conjugating enzyme Ube2l2, and Lrrk2 itself (Fig. EV5D). Correlation analyses mirrored the kidney, showing opposing alterations between KO and G2019S, with both inhibitors positively correlating with KO and strongly with each other (Appendix Fig. S13D). Pathway enrichment highlighted neuronal compartments—dendrites and axons—as significantly affected by Lrrk2 perturbations (Fig. EV5E). Twenty-four proteins altered in the brain were also changed in human urine from the PPMI cohort (Fig. EV5F), supporting the relevance of urinary proteomics for capturing central nervous system (CNS)-linked molecular changes.

Together, these findings demonstrate tissue-specific proteomic signatures of Lrrk2 perturbations, with the kidney showing a stronger response among the tissues examined and the largest overlap of significantly regulated proteins with urine, as expected (Appendix Fig. S12E). Notably, urine exhibited an even stronger aggregated effect than the corresponding tissues (Appendix Fig. S12F). G2019S and KO drive opposing proteomic changes, and kinase inhibition largely recapitulates loss-of-function states. Importantly, cross-species overlap between rat tissues and human urine underscores the translational value of urinary proteins as non-invasive biomarkers of LRRK2 perturbations across peripheral organs and the CNS.

## Discussion

Here, we present a comprehensive urinary proteomics resource in PD, combining three human cohorts (PPMI, LCC, Columbia) and quantifying 7663 proteins with a median analytical CV of 25%. Using a systems-biology framework to connect genotype to biofluid proteome, we integrated whole-genome sequencing from >500 PPMI participants (58 variants across 45 loci, including multiple LRRK2 and GBA mutations) to map how PD risk variants shape the urinary proteome. This revealed both convergent and genotype-specific urinary signatures, with LRRK2—especially G2019S, followed by R1441G—exerting the strongest effects. The smaller effect size for R1441G relative to G2019S likely reflects both the smaller cohort (26 carriers vs. 305 G2019S carriers in PPMI) and potentially a distinct biochemical mechanism: G2019S directly hyperactivates the kinase domain, whereas R1441G impairs ROC domain GTP hydrolysis, leading to more indirect kinase hyperactivation (Schmidt et al, 2019; Wu et al, 2019). Larger,

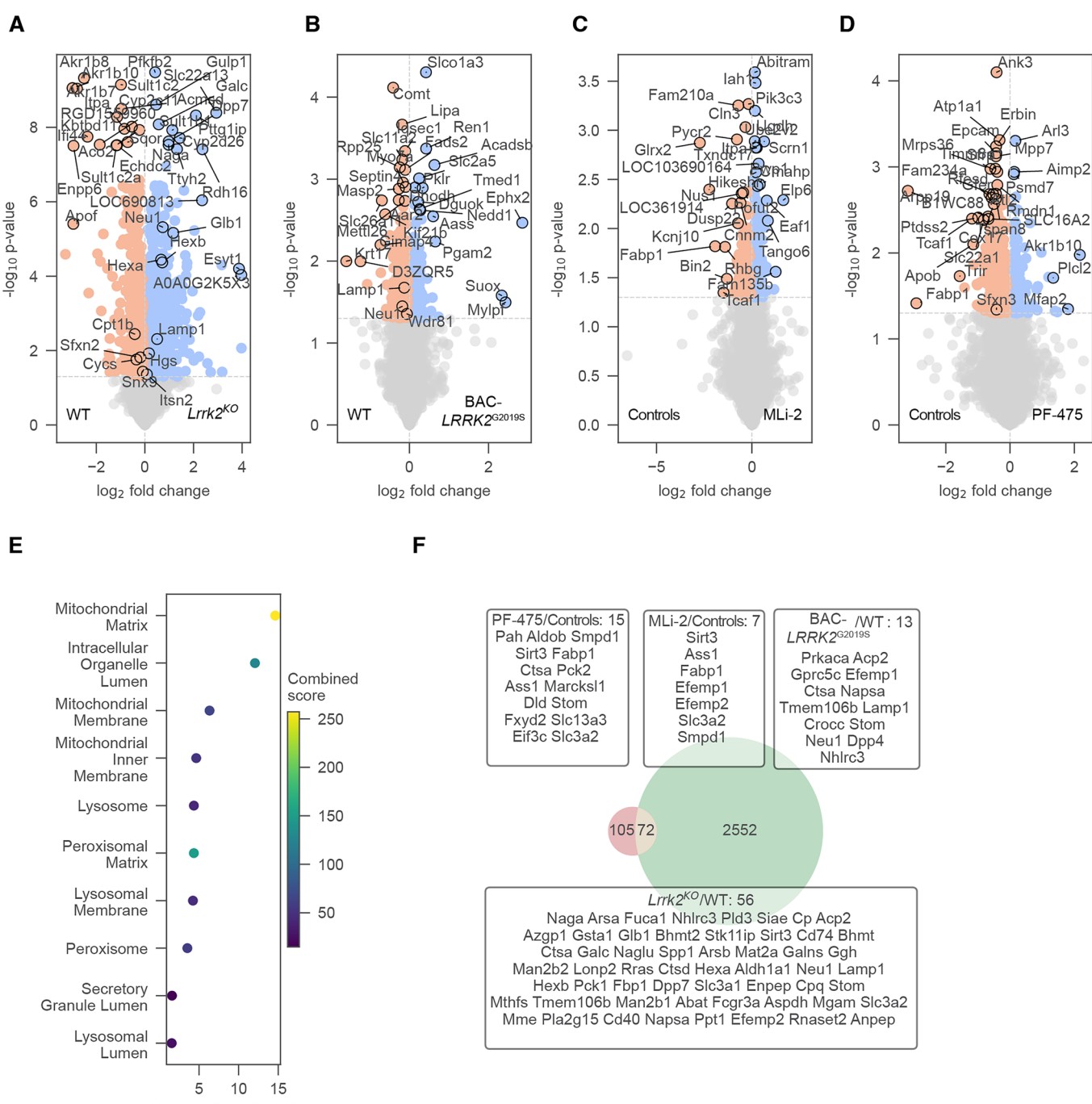

**Figure 5. Detecting proteome alterations associated with Lrrk2 loss, hyperactivation (G2019S), and kinase inhibition in rat kidney.**

(**A–D**) Volcano plots comparing the kidney proteomes of $Lrrk2^{KO}$ vs. $Lrrk2^{WT}$ rats ($n = 8150$) (**A**), BAC-$LRRK2^{G2019S}$ vs. $Lrrk2^{WT}$ rats ($n = 8009$) (**B**), MLi-2 vs. Controls ($n = 7320$) (**C**), and PF-475 vs. Controls ($n = 7320$) (**D**) with dotted lines indicating 5% $p$ value cutoffs. The vertical dotted line separates the proteins into up- and down-regulation. Most relevant proteins are annotated. (**E**) Enrichment analysis of the GO term "cellular component". Enrichment was performed on significant proteins ($p$ value <5%) in at least two out of four comparisons. The combined score represents an aggregated metric calculated during enrichment analysis. (**F**) Venn diagram of union of significant proteins ($p$ value <5%) in $Lrrk2^{KO}$ vs. $Lrrk2^{WT}$ or BAC-$LRRK2^{G2019S}$ vs. $Lrrk2^{WT}$ or MLi-2 vs. Controls or PF-475 vs. Controls (right) and network genes from human study (left). Intersecting proteins are annotated in a box with their respective comparison.

independent genotype-specific cohorts will be needed to clarify and validate these differences.

Beyond R1441G, *COMT* variants including rs165656 (Zhong et al, 2025) and rs4633 (Bialecka et al, 2008; Qian et al, 2017) also exhibited G2019S-like urinary proteomic patterns. *COMT* encodes catechol-O-methyltransferase, which exists as membrane-bound (enriched in the brain) and soluble (abundant peripherally, such as in the liver and kidney) isoforms that methylate catecholamines (e.g., dopamine and norepinephrine). Clinically, COMT inhibitors are used with levodopa to reduce peripheral metabolism and improve CNS delivery (Kaakkola, 2000). A plausible explanation for the COMT–LRRK2$^{G2019S}$ convergence is indirect and requires functional testing: risk alleles such as rs165599, located in the 3′ UTR, can lower COMT expression via altered mRNA stability/ miRNA binding (Dempster et al, 2006). With lower COMT, catecholamines are cleared less efficiently peripherally, producing reactive breakdown products that stress the autophagy–lysosome system. This can disrupt glycosphingolipid turnover and lysosomal hydrolase trafficking-pathways enriched in our urinary signature (e.g., cathepsins, HEXB, NEU1, and GM2A).

Strikingly, *GBA* mutations (both E365K and N409S) showed inverse regulation with *LRRK2* variants. We previously observed this pattern in a smaller cohort (Virreira Winter et al, 2021); the larger sample size here enables confirmation of these associations. *GBA* encodes glucocerebrosidase (GCase), a key lysosomal enzyme in sphingolipid metabolism, and heterozygous *GBA* variants increase PD risk, whereas biallelic loss causes Gaucher disease (Smith et al, 2017). We compared *GBA*- and *LRRK2*-associated profiles in depth, and our analyses supported that both involve lysosomal/glycosphingolipid pathways with mutation-specific directionality between the groups, consistent with divergent mechanisms converging on shared disease biology. These inverse effects are consistent with human data showing elevated peripheral Gcase activity in *LRRK2*$^{G2019S}$ carriers (Alcalay et al, 2015, 2018) and increased activities of other lysosomal hydrolases in the same genotype, contrasted with the partial Gcase loss-of-function typical of *GBA*$^{E365K/N409S}$ carriers (Alcalay et al, 2015; Huh et al, 2023). Clinically, patients carrying both *LRRK2* and *GBA* variants show milder progression, particularly slower cognitive decline, than *GBA*-only PD patients, with further support from a recent Drosophila model in which mutant LRRK2 attenuated GBA1-linked phenotypes (Omer et al, 2020; Ortega et al, 2021; Serebryany-Piavsky et al, 2025; Yahalom et al, 2019). Together, these observations are consistent with antagonistic or modifying interactions between LRRK2 hyperactivity and GBA-related deficits. Future studies should test whether GCase-boosting therapies normalize proteins inverse to the LRRK2 signature and restore pathway balance.

To evaluate translational relevance, we combined human data with *Lrrk2*$^{KO}$, mutant, and inhibitor-treated rat urine studies. Using machine learning, we derived a LRRK2-associated urinary protein signature—including core lysosomal proteins such as cathepsins, GM2A, PLD3, HEXB, and NEU1—that generalizes across cohorts and is conserved in rats and modulated by LRRK2 kinase inhibition. These results support its potential as a robust pharmacodynamic readout and establish the rat urinary proteome as a causal engine for biomarker discovery and validation. Shifting the clinical focus from "target engagement" toward "pathway normalization" would enable a more comprehensive assessment of

therapeutic efficacy in rescuing the systems-level defects that characterize LRRK2-driven pathology. Furthermore, this urinary signature identified individuals without G2019S who exhibited "G2019S-like" profiles, including carriers of other *LRRK2* variants like N2081D, which overlaps with inflammatory bowel disease phenotypes or GBA variants and some with idiopathic PD. This supports leveraging biology-based and pathway-centric, rather than genotype-only, stratification of both mutation-positive and -negative PD patients for LRRK2-targeted therapies.

Two observations merit further investigation. First, while *Lrrk2*$^{KO}$ and kinase inhibition produced the expected inverse proteomic patterns relative to BAC-*LRRK2*$^{G2019S}$, we noted a modest positive correlation between G2019S and MLi-2 in the aggregate analysis of rat urine. Most panel proteins shifted in the opposite direction as expected, suggesting that the positively correlating subset may reflect compound-specific effects, sampling time-points/dose or adaptive responses. Time-course and dose-response studies with both Type I and Type II inhibitors would be informative. Second, several top markers correlated with clinical severity in *LRRK2* PD—but not in non-*LRRK2* PD—raising the possibility of subtype-specific progression markers that warrant longitudinal validation for prognostic value and responsiveness to LRRK2 inhibition.

Existing LRRK2-pathway biomarkers—phosphorylation readouts (Rab10 Thr73 as well as LRRK2 Ser935 and Ser1292 phosphorylations) and urinary BMPs (notably di-22:6-BMP)—are proven pharmacodynamic tools for confirming target engagement and guiding dose in trials (Alcalay et al, 2020; Gomes et al, 2023; Ö. Karayel et al, 2020; Taymans, Mutez, et al, 2023). However, each has constraints: BMPs primarily index lysosomal lipid handling and are not uniformly elevated across genotypes, and phosphorylations are often subject to substoichiometric limitations and high technical variability in biofluids. LRRK2 Ser1292 phosphorylation is particularly challenging to detect as it is substoichiometric and especially by mass spectrometry, as trypsinization yields a very short peptide (Rideout et al, 2020). In the unpublished targeted urinary proteomics analysis of the PPMI cohort, total RAB10 and LRRK2 seemed to be elevated in G2019S carriers while phosphosite signals were sparse, consistent with detection limits rather than absence of biology (Appendix Fig. S15). Nevertheless, the regulation of total protein levels in urine was consistent with our global proteomics analysis, which also demonstrated increased RAB10 (coefficient 0.48, q = 2.6e-5) and LRRK2 (coefficient 0.63, q = 2e-6) in G2019S urine (EV3B; Dataset EV2). These findings align with previous urinary exosomal vesicle studies (Fraser et al, 2016; Taymans, Mutez, et al, 2023), which also reported increased RAB8 and RAB10 abundance but unchanged phosphorylation rates in patient urinary extracellular vesicles (EVs) (Taymans, Mutez, et al, 2023). While phosphorylation-based markers remain essential for confirming target engagement, our multiprotein panel will complement these single-analyte markers, providing a more reliable and resilient readout that reports multi-pathway consequences (lysosome, glycosphingolipid, immune, and trafficking) of LRRK2 dysfunction and inhibition. We advocate for a tiered biomarker strategy: clinical trials utilizing BMPs/ pRab10 as primary target-engagement markers while leveraging our multiprotein panel for longitudinal monitoring of "biological rescue." This framework moves beyond binary readouts toward a granular, pathway-level understanding of dose-response and therapeutic efficacy. We need future work focused on assay harmonization and targeted verification to enable clinical translation of the panel proteins.

While these findings establish urine as a valuable, scalable biofluid for discovering pathway-centric biomarkers, the extent to which neurodegenerative processes are mirrored in the urinary proteome should be explored further. Cross-biofluid comparisons (O. Karayel et al, 2022; Rutledge et al, 2024) show only partial overlap with brain-proximal matrix cerebrospinal fluid (CSF) and urine. Accordingly, matched deep biofluid (such as CSF and plasma) proteomes from the same individuals will be essential to delineate shared versus compartment-specific signals and to validate urinary readouts of systemic changes and brain-relevant biology. To contextualize urinary signals at the tissue level, we also performed proteomics on proximal (kidney) and distal (lung, brain) tissues in Lrrk2 rat models. This analysis revealed the strongest phenotypic similarity shared with the kidney (278 proteins) (Appendix Fig. S12E), consistent with high LRRK2 expression in renal tissue, particularly in proximal tubule cells (Baptista et al, 2013; Kluss et al, 2021; Thévenet et al, 2011). It is therefore not surprising that among the 58 mutations across 45 loci we evaluated, *LRRK2* mutations exerted the most pronounced effects on the urinary proteome, providing a unique opportunity to interrogate LRRK2 dysfunction non-invasively. Yet, lung (137) and brain (117) signals were also detectable in rat urine (Appendix Fig. S12E).

Furthermore, the identification of 24 proteins in human urine that were also altered in the rat brain reinforces the capacity of the urinary proteome to capture signals from distal organs, establishing it as a non-invasive proxy matrix for brain-relevant molecular changes. Notably, aggregate fold-changes were often more pronounced in urine than in individual tissues (Appendix Fig. S12F), suggesting that urine acts as a sensitive integrator of LRRK2-linked biology across compartments. This sensitivity may reflect both filtration/enrichment and EV shedding dynamics and merits mechanistic studies in the future. In late-stage clinical trials, where high-frequency longitudinal monitoring is essential to assess dose-response and long-term safety, these urinary readouts offer a scalable, low-burden alternative to invasive CSF collection.

In sum, our study establishes urine as a scalable, non-invasive biofluid for the proteomic monitoring of LRRK2 dysfunction in PD. It captures both kidney-proximal and brain-relevant consequences of LRRK2 dysfunction, offering a non-invasive window into CNS-linked biology. Furthermore, it bridges genetic and idiopathic PD through pathway-centric signatures that enable biology-based patient stratification and identify a multiprotein panel for pharmacodynamic monitoring to assess pathway normalization. These findings deliver a high-fidelity toolkit to directly support the ongoing and future clinical development of LRRK2-targeted therapies.

## Methods

### Reagents and tools table

| Reagent/resource | Reference or source | Identifier or catalog number |
|---|---|---|
| **Experimental models** | | |
| Wistar rat | Janvier Labs | Rj:WI |
| Long Evans Hooded rat | Horizon Discovery (currently Inotiv) | HsdBlu:LE |

| Reagent/resource | Reference or source | Identifier or catalog number |
|---|---|---|
| LRRK2 KO rat | Horizon Discovery (currently Inotiv) | Lrrk2 KO (LE-Lrrk2tm1sage) |
| BAC-LRRK2-G2019S rat | Oxford University | Prof. Richard Wade-Martins (Sloan et al 2016) |
| **Antibodies** | | |
| Anti-total LRRK2 | UC Davis/NIH NeuroMab Facility | 75-253 |
| Anti-pS935-LRRK2 | Abcam | ab133450 |
| Anti-alpha-tubulin | Sigma-Aldrich | T9026 |
| Anti-beta-actin | Sigma-Aldrich | A5441 |
| Anti-rabbit HRP | Cell Signaling | 7074S |
| Anti-mouse HRP | Cell Signaling | 7076S |
| **Chemicals, enzymes and other reagents** | | |
| Urea | Sigma-Aldrich | 57-13-6 |
| Ammonium bicarbonate | Sigma-Aldrich | 1066-33-7 |
| DL-Dithiothreitol | Sigma-Aldrich | 3483-12-3 |
| Iodoacetamide | Sigma-Aldrich | 144-48-9 |
| 96-well PVDF membrane plates | Merck | MSIPS4510 |
| Trypsin Protease | Sigma-Aldrich | T6567 |
| LysC Protease | Sigma-Aldrich | LYSC9000 |
| Acetonitril | Sigma-Aldrich | 34851 |
| Ethanol | Sigma-Aldrich | E7023 |
| Formic Acid | Sigma-Aldrich | 06473 |
| Evosep Tips | Evosep | EV2011 |
| Tris Base | Sigma-Aldrich | 77-86-1 |
| EGTA | Sigma-Aldrich | 60-00-4 |
| Formic acid | Sigma- Aldrich | 64-18-6 |
| MLi-2 | R&D Systems | 5756 |
| PF-06447475 (PFE-475) | Selleckchem | S8202 |
| Pierce™ BCA protein assay | Thermo Fisher | 23225 |
| Nitrocellulose membrane | GE Healthcare | 10401197 |
| NuPAGE TM NOVEX TM 3–8% Tris-Acetate | Thermo Fisher Scientific | TA03812BOX |
| NuPAGE TM NOVEX TM 4–12% Bis-Tris | Thermo Fisher Scientific | NP0322BOX |
| NuPAGE TM NOVEX TM 4–20% Tris-Glycine | Thermo Fisher Scientific | XP04202BOX |
| NuPAGE TM NOVEX TM 12.5% | Thermo Fisher Scientific | NP0342BOX |
| Amsersham ECL prime | Cytiva | RPN2236 |
| **Software** | | |
| DIA-NN v1.8.1 | Demichev et al, 2020 | |
| **Other** | | |
| Orbitrap Exploris 480 | Thermo Fisher Scientific | |

| Reagent/resource | Reference or source | Identifier or catalog number |
|---|---|---|
| Bravo Automated Liquid Handling Platform | Agilent | |
| Nanodrop 2000 | Thermo Fisher Scientific | |
| Evosep One HPLC | Evosep | |
| timsTOF Pro 2 | Bruker | |
| Pepsep column | Bruker | |
| EASY-nLC 1200 system | Thermo Fisher Scientific | |

## Study cohorts

The Parkinson's Progression Markers Initiative (PPMI) is a global observational study aimed at identifying biomarkers for Parkinson's disease (PD) risk. Operating across Europe, Israel, Canada, and Africa, the initiative has enrolled over 4000 participants who undergo regular clinical assessments, imaging, bio sample collection, and genetic analysis. Our research focuses on PPMI Project 190, which includes three groups: 477 PD patients with early untreated PD or genetic *LRRK2* and *GBA* variants, 362 prodromal participants at risk of developing PD based on clinical features, genetic variants, or other biomarkers, and 141 healthy controls with no neurological disorders or family history of PD (Appendix Fig. S16). Proteomics and metadata for this study were obtained from the PPMI database in July 2023 (http://www.ppmi-info.org). For the latest updates, visit http://www.ppmi-info.org.

The Columbia cohort, established at Columbia University Irving Medical Center as part of the MJFF-funded LRRK2 biomarker initiative, consists of 120 participants enrolled between March 2016 and April 2017. This cohort includes 35 healthy controls without *LRRK2* mutations, 16 asymptomatic *LRRK2* G2019S mutation carriers, 40 idiopathic PD (iPD) patients, 28 PD patients with the *LRRK2* G2019S mutation, and one PD patient with an undetermined *LRRK2* mutation status. The LCC cohort includes 115 bio banked urine samples, also part of the MJFF-funded LRRK2 Cohort Consortium, comprising 26 healthy controls, 37 asymptomatic *LRRK2*[G2019S] mutation carriers, 29 idiopathic PD patients, and 23 PD patients with the *LRRK2*[G2019S] mutation.

## Quality assessment for sample collection and data generation of the PPMI cohort

Detailed patient recruitment and sample collection protocols are available on the official PPMI website (https://www.ppmi-info.org/study-design/research-documents-and-sops).

## Patient recruitment

PPMI employs a comprehensive multi-cohort recruitment strategy encompassing three main groups: (1) PD patients with confirmed diagnoses, including carriers of genetic variants (*LRRK2*, *GBA*, and *SNCA*); (2) prodromal participants at high risk for PD but without motor symptoms, identified through REM sleep behavior disorder, olfactory loss, genetic risk variants, or other established risk factors; and (3) healthy controls. Recruitment is conducted through a multicenter international network utilizing both traditional clinical enrollment and digital recruitment pathways, with centralized coordination for genetic mutation carriers.

## Urine sample collection and processing

Urine samples were collected and processed following standardized protocols to ensure sample integrity. Midstream urine (minimum 7.5 mL) was collected in labeled tubes and centrifuged at 4 °C for 15 min at 2500×*g* within 30 min of collection to remove cellular debris. The clarified supernatant was transferred to clean tubes and stored at −80 °C within 60 min of collection, with all processing times documented in the study database.

## Data generation and quality control

To ensure data reliability and reproducibility, we implemented systematic quality control measures during proteomic analysis. Pooled reference samples were positioned at every 12th well (A12-H12) on each 96-well plate. Sample randomization across plates and analytical runs was performed to distribute healthy controls and disease samples evenly, thereby minimizing batch effects and ensuring that observed variations represent true biological differences rather than technical artifacts.

## PPMI urine sample preparation and LC-MS/MS analysis

Neat urine from the PPMI cohort was prepared using the MStern Blot protocol (Berger et al, 2015) adapted to semi-automated processing on an Agilent Bravo Robot. Briefly, 100 µl of urine were diluted in 300 µl of urea sample solution (8 M urea in 50 mM ammonium bicarbonate (ABC)) and subsequently mixed with 30 µl of 150 mM dithiothreitol (DTT) solution (150 mM DTT, 8 M urea, and 50 mM ABC) in a 96-well plate. The resulting solution was incubated for 20 min at room temperature. Reduced cysteine side chains were alkylated by adding 30 µL of iodoacetamide (IAA) solution (700 mM IAA, 8 M urea, and 50 mM ABC) and incubated for 20 min. During incubation, each well of the 96-well PVDF membrane plates (MSIPS4510, Merck Millipore) was activated and equilibrated with 150 µl of 70% ethanol/water and urea sample solution, respectively. The urine samples were transferred through the PVDF membranes using the Agilent Bravo vacuum filtration station. Adsorbed proteins were washed twice with 150 µl of 50 mM ABC. Digestion was performed at 37 °C for 4 h by adding 100 µl digestion buffer (5% v/v acetonitrile (ACN)/50 mM ABC) containing 0.35 µg per well of each protease trypsin and LysC. After incubation in an incubator, the resulting peptides were collected by applying vacuum and the remaining peptides were eluted twice with 75 µl of 40%/0.1%/59.9% (v/v) acetonitrile/formic acid/water. The pooled peptide solutions were dried in a vacuum centrifuge. Peptides were resuspended in 20 µl buffer A (0.1% formic acid in water), and peptide concentrations were measured optically at 280 nm (Nanodrop 2000, Thermo Scientific) and subsequently equalized using buffer A. 500 ng peptide were loaded on Evosep Tips for LC-MS/MS analysis according to the manufacturer's protocol.

## Data independent acquisition of PPMI urine samples

LC-MS/MS analysis was performed on an Evosep One HPLC coupled to a timsTOF Pro 2 mass spectrometer. Peptides were separated at 60 °C on a 15 cm PepSep column with an inner diameter of 150 μm, coupled to a 20 μm Bruker ZDF sprayer on the Evosep 30 SPD gradient. MS data were acquired in diaPASEF mode at 100 ms ramp time.

## Animal study

All animal procedures were approved by the Ethical Committee of the French Ministry of National Education, Higher Education and Research (reference #4271-2015102712365542). Rodent housing and handling of rodents were done in compliance with national and international guidelines. Wistar rats (Janvier Labs, Le Genest-Saint-Isle, France) were used in the pharmacological treatment experiments.

Pharmacological treatment of rats was done via intraperitoneal injection. Injected doses in the rats are 30 mg/kg for PFE-475 and 2 mg/kg for MLi-2. *Lrrk2*[KO] rats (HsdSage: LE-Lrrk[tm1sage]) (Baptista et al, 2013) and wild-type Long Evans hooded rats were ordered from Horizon Discovery (currently, Envigo, Lafayette, CO, USA). Finally, the BAC-*LRRK2*[G2019S] transgenic rats were obtained from Prof. Richard Wade-Martins (Oxford University), using littermate non-transgenic rats as controls (Sloan et al, 2016).

LRRK2 kinase inhibitors: Compounds used for pharmacological treatments are PF-06447475 (PFE-475 for short, Selleckchem, Houston, TX, USA) and MLi-2 (R&D Systems, Abingdon, UK).

## Rat urine collection and preparation

Rat urines were collected over a 7–8-h period in metabolic cages. In these cages, urine produced by the rodents is sorted into urine collection vessels that are prepared to contain 100 μL of Tris-EGTA buffer (1 M Tris pH 7.4, and 40 mM EGTA) at the beginning of collection. The urine collection vessels are placed in a cooling module for the duration of the collection to ensure that the collected urine is maintained cool. At the end of the collection period, urines are complemented with Tris-EGTA to a final concentration of 5% and sampled into 1.5 mL aliquots. Urine is then stored at −80 °C until processing. Neat rat urine preparation was done the same way as in the PPMI cohort.

## Rat tissue collection and preparation

Rats were anesthetized with ketamine (100 mg/kg)/xylazine (10 mg/kg) and then transcardially perfused with HBSS. Tissues, including the brain, kidney, and lungs, were rapidly dissected out and snap frozen using liquid nitrogen. Tissues were stored at −80 °C until use.

To make homogenates for use in western immunoblotting, tissues were ground with a mortar and pestle in liquid nitrogen and homogenized in 5 volumes of buffer medium (10 mM Tris-HCl, 1 mM EDTA, and 0.25 M sucrose, pH 7.4), containing a Complete® protease inhibitor cocktail (Roche Molecular Biochemicals, Indianapolis, IN, USA), using a Dounce homogenizer. An aliquot of the resultant homogenates was stored at −80 °C.

Samples were resuspended in 100 mM Tris-HCl, pH 8.5, in 1% (w/v) sodium deoxycholate (SDC) and alkylated and reduced with 10 mM Tris(2-carboxyethyl)phosphine (TCEP), 40 mM 2-chloracetamide (CAA) at 45 °C for 5 min. Proteins were then digested using a 1:1000 ratio of trypsin and LysC:protein at 37 °C overnight with 1200 rpm agitation on an Eppendorf Thermomixer C. The next day, samples were ten-fold diluted using 1% trifluoroacetic acid (TFA) in isopropanol. Desalting was done using SDB-RPS (Empore) StageTips. For that, samples were loaded on StageTips and subsequently washed twice with 200 μL of 1% TFA in isopropanol and twice with 0.2% TFA/2% acetonitrile (ACN). Peptides were eluted using 80% ACN/1.25% NH4OH and dried by SpeedVac centrifuge (Concentrator Plus; Eppendorf) for 1 h at 30 °C. Peptides were then resuspended in 0.2% TFA/2% ACN and analyzed using LC-MS/MS.

## Data independent acquisition of rat urine and tissue samples

We used the following LC-MS/MS set up for a data independent acquisition (DIA): Peptides were eluted on a 50 cm reversed-phase column (75-μm inner diameter, packed in-house with ReproSil-Pur C18-AQ 1.9 μm resin) with an EASY-nLC 1200 system from Thermo Fisher Scientific, with a column temperature of 50 °C maintained using an in house made column oven. MS/MS analysis was performed using an Orbitrap Exploris 480 from Thermo Fisher Scientific via a nano-electrospray source. Peptides were separated using a binary buffer system consisting of buffer A (0.1% formic acid (FA)) and buffer B (80% ACN, 0.1% FA) with a constant flow rate at 300 nL/min and a gradient length of 75 min. The gradient starts with 2% buffer B and gradually increases to 35% after 60 min. 60% after 70 min and reaches its maximum at 90% after 71 min where it remains constant until 75 min. We used an MS set-up as described before (Steger et al, 2021). We used a DIA mode with the following settings: full scan range: 300–1650 m/z at 120,000 resolution, automatic gain control: 3e6, maximum injection time: 20 ms, stepped higher-energy collision dissociation (HCD): 25, 27.5, and 30, number of DIA scans: 44 at 30,000 resolution, AGC: 1e6 and maximum injection time: 54 ms.

## Linear regression and network analysis

To estimate significant regulation in our urine PPMI dataset, a multiple linear regression (MLR) was trained on standardized confounders. Using a multiple linear regression model for each protein, we accounted for confounders including age and sex. An example equation for the MLR looks as follows:

$$y = \beta_0 + \beta_1 X_1 + \beta_2 X_2 + \beta_3 X_3 + \ldots + \beta_n X_n$$

Where y is the protein intensity, $\beta_0$ the intercept, $\beta_1$ and $\beta_2$ the coefficients for age ($X_1$) and sex ($X_2$) and $\beta_{3n}$ are the coefficients for the n-2 remaining variables ($X_{3n}$), such as the 58 risk variants or specific mutation indicators. The coefficients and their corresponding *p*- and *q*-values were extracted for each protein for each confounder. For the *q*-value calculation, we used the Benjamini–Hochberg method for multiple testing correction. A network was created based on significant proteins (*q*-value <5%)

from G2019S or R1441G. For each pair of significant proteins, a separate ridge regression model was trained using one protein's $\log_2$ intensity to predict the other's $\log_2$ intensity, resulting in a coefficient, $p$ value, $q$ value and $R^2$ adjacency matrix. We considered a protein-protein interaction only if a pair of proteins had a $q$ value <5% and an absolute coefficient of at least 0.15. We used the Louvain clustering algorithm to separate the network into individual functional clusters. To estimate the most pivotal elements in the network, proteins were ranked by their $R^2$ value to predict the $\log_2$ intensity of all proteins. Using the elbow method, we estimated when model performance based on the $R^2$ value stagnates as we incrementally increase the number of proteins to predict the $\log_2$ intensity of the entire dataset. Incremental addition was done with the next-best protein based on its $R^2$ rank. We obtained WGS data from 534 patients in the urinary PPMI cohort (downloaded August 2023 from https://www.ppmi-info.org). The WGS dataset contains single-nucleotide polymorphism (SNP) identifiers encoded as follows: homozygous mutations are represented as 2, heterozygous mutations as 1, and absence of mutation as 0. We employed two analytical approaches. For the aggregate analysis, we grouped mutations affecting the same gene locus by summing their values, then performed linear regression analysis. Alternatively, we conducted a non-aggregated analysis using the raw WGS data without prior aggregation. To obtain comprehensive *LRRK2* and *GBA* mutation status information, we additionally utilized the genetics consensus file (downloaded August 2023 from https://www.ppmi-info.org/), which provides detailed genetic status for all 980 patients in the urinary study cohort.

## Machine learning

The LCC, Columbia and PPMI datasets were first split into a train (70%) and a test dataset (30%). The train datasets from each cohort were combined for feature selection and training. Using a combination of mutual information and the $F$-test, we first reduced the number of features in the training data. The reduced set of features were then ranked based on its absolute coefficient using an SVM classifier with a linear kernel. To estimate the optimal number of features, we incrementally added the next-best-ranked features to our model and monitored ROC AUC, accuracy, and average precision on the training and validation datasets using a repeated stratified fold with five splits and three repeats. The optimal number of features was determined at the intersection between model performance on the training and validation to prevent overfitting. An SVM classifier was then trained on the reduced training data and tested on a separate PPMI, LCC and Columbia test dataset. For the cross-cohort model, we applied a similar approach with the difference that the feature selection was first done on the entire PPMI training data, and selected features were then removed if they were not identified in the LCC or Columbia test data to prevent data leakage before feature selection. Estimation of the optimal numbers of features and model performance evaluation was done the same way as described above. For the batch correction, the LCC, Columbia and PPMI cohort was first combined and subsequently batch-corrected using the inmoose v0.7.6 package (Behdenna, 2023). The same pipeline mentioned above was then used for classification and model performance evaluation.

## DIA data processing and bioinformatics analysis

DIA data were processed using DIA-NN v1.8.1. (Demichev et al, 2020) against a *Rattus norvegicus* and human Uniprot proteome FASTA file with canonical and isoform sequences. The library-free search was performed with FASTA digest for library-free search and the deep learning-based spectra, RTs, and IMs (retention time and ion mobility) predictions enabled and heuristic protein inference activated. We included carbamidomethylation, oxidation of methionine, and N-terminal acetylation of proteins as modifications. The remaining settings were left at default. The bioinformatics analysis was done using Python 3.11.3 with the following packages: pandas 1.4.2, numpy 1.21.5, matplotlib 3.5.13, seaborn 0.11.2, statsmodels 0.13.5, sklearn 1.4.2, and scipy 1.14.1. For the rat tissue proteomics datasets, intensities were first $\log_2$-transformed to attain data normality. Samples were deemed outliers by counting the number of proteins with an absolute robust $z$-score above 3.5 (outlier proteins) and a total outlier protein frequency above 1.5 x IQR across each experimental group. We then filtered the data for valid values in at least one experimental group and imputed missing values using a sampling method from a shifted Gaussian distribution (shift: 3 standard deviations, width: 0.3 standard deviations). In order to determine statistical significance, we used an unpaired two-tailed Student's $t$-test and adjusted $p$ values using the Benjamini–Hochberg method. For the PPMI, Columbia, and LCC dataset we used the same preprocessing steps as mentioned before, with the exception of protein filtering and imputation. Blood contamination outliers were removed with the strategy described earlier (Geyer et al, 2019). Outliers from the Columbia and LCC cohort were adapted from previous study (Virreira Winter et al, 2021).

## Western blotting

Protein content of cell lysates was determined using the bicinchoninic acid (BCA) protein determination assay (Pierce Biotechnology) or the Bradford method (Thermo Scientific) with bovine serum albumin (BSA) as the standard. After addition of LDS sample buffer (containing sample reducing agent) and boiling, equal volumes of samples (10 µl) were resolved by electrophoresis on NuPAGE 3–8% Tris-Acetate gradient gels, 4–12% Bis-Tris gradient gels, 4–20% Tris-Glycine gradient gels, or 12.5% SDS gels (LifeTechnologies). Separated proteins were transferred to PVDF (Bio-Rad) or nitrocellulose (Amersham) membranes, and non-specific binding sites were blocked for 60 min in Tris-buffered saline containing 0.05% Tween-20 (TNT) and 5% non-fat milk or 5% BSA. After overnight incubation at 4 °C with the appropriate antibodies, blots were washed four times with TNT. After incubation with the secondary antibodies, blots were washed again. Bands were visualized using enhanced chemiluminescence (Amersham ECL Prime, Cytiva) that was acquired using the image analyzer Imager 600 (GE Healthcare Bio-Sciences). All blots were processed in parallel and derived from the same experiment.

# Data availability

The rat proteomics data have been deposited to the ProteomeXchange Consortium via the PRIDE partner repository with the dataset

identifier PXD057308. The proteomics data from the LCC and Columbia cohorts have been previously published (Virreira Winter et al, 2021) and are available in the ProteomeXchange Consortium via the PRIDE partner repository with the dataset identifier PXD020722. The proteomics and clinical metadata from the PPMI cohort used in this study are hosted by the PPMI database and are available for download at http://www.ppmi-info.org.

The source data of this paper are collected in the following database record: biostudies:S-SCDT-10_1038-S44320-026-00190-0.

## Peer review information

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

## Acknowledgements

The data used in this study were obtained from MJFF-sponsored cohorts. We thank the Proteomics and Signal Transduction Group at the Max Planck Institute of Biochemistry, the Clinical Proteomics Group at the NNF Center for Protein Research, especially Igor Paron for technical assistance, and Dario Alessi and Esther Sammler from the University of Dundee, Sarah Huntwork-Rodriguez from Denali Therapeutics as well as employees of the Michael J. Fox Foundation for Parkinson's Research, in particular Shalini Padmanabhan and Samantha Hutten, for their helpful discussions. This project was supported by the Michael J. Fox Foundation (MJFF-12938.4 and MJFF-004693) and by the Max Planck Society for the Advancement of Science. AF-G and LG received scholarships from the Université de Lille (Ecole Doctorale Biologie-Santé).

## Author contributions

**Duc Tung Vu**: Conceptualization; Formal analysis; Investigation; Visualization; Methodology; Writing—original draft; Writing—review and editing. **William Sibran**: Investigation. **Andreas Metousis**: Investigation; Methodology; Writing—review and editing. **Laurine Vandewynckel**: Investigation. **Basak Eraslan**: Formal analysis; Methodology. **Liesel Goveas**: Investigation. **Ericka CM Itang**: Investigation; Methodology; Writing—review and editing. **Claire Deldycke**: Investigation. **Adriana Figueroa-Garcia**: Investigation. **Réginald Lefèbvre**: Investigation. **Johannes Bruno Müller-Reif**: Conceptualization; Investigation; Methodology; Writing—review and editing. **Sebastian Virreira-Winter** Conceptualization; Formal analysis; Funding acquisition; Investigation; Methodology; Project administration; Writing—review and editing. **Marie-Christine Chartier-Harlin**: Funding acquisition; Writing—original draft; Conceptualization; Supervision; Project administration; Writing—review and editing. **Jean-Marc Taymans**: Conceptualization; Supervision; Funding acquisition; Writing—original draft; Project administration; Writing—review and editing. **Matthias Mann**: Conceptualization; Supervision; Funding acquisition; Project administration; Writing—review and editing. **Ozge Karayel**: Conceptualization; Formal analysis; Supervision; Investigation; Visualization; Methodology; Writing—original draft; Project administration; Writing—review and editing.

Source data underlying figure panels in this paper may have individual authorship assigned. Where available, figure panel/source data authorship is listed in the following database record: biostudies:S-SCDT-10_1038-S44320-026-00190-0.

## Funding

## Disclosure and competing interests statement

The authors disclose the following activities and funding sources not directly related to this manuscript: MCCH has served on advisory boards for the European League Against Alzheimer's Disease (Ligue Européenne contre la maladie d'Alzheimer) and currently serves on the Board of Directors for the Foundation Vaincre Alzheimer and its Scientific Development Committee; these roles involved no financial compensation. J-MT and M-CC-H have received research grants from the Michael J. Fox Foundation, the European Commission, France Parkinson, and the Agence Nationale de la Recherche (ANR). J-MT also received funding from the European Regional Development Fund. MM serves as an editorial advisory board member; this role has no bearing on the editorial consideration of this manuscript. SVW is associated with ions.bio. OK is currently affiliated with Genentech. The remaining authors declare no competing interests.

# Expanded View Figures

**Figure EV1.  Detecting urinary proteome alterations associated with genetic risk variants in PD.**

(**A**) Number of significant proteins for each genetic risk variant based on multiple linear regression analysis on WGS data. Each genetic variant is described by gene name followed by the unique NCBI dbSNP database identifier. (**B**) Pearson correlation coefficient and corresponding $-\log_{10} q$ values of beta-value profile derived from multiple linear regression analysis between $LRRK2^{G2019S}$ and other genetic risk variants. (**C**, **D**) Reactome enrichment plot of significant proteins ($q$ value <1% and |beta|>0.2) from linear regression analysis for G2019S and R1441G. The overlap illustrates the proportion of enriched proteins in each term, with the most relevant terms annotated. The combined score represents an aggregated metric calculated during enrichment analysis. (**E**) Volcano plots showing the relationship between $-\log_{10} p$ values and beta values for Prodromal, PD $LRRK2^{G2019S}$, PD, $GBA^{E365A}$, and $GBA^{N409S}$ ($n = 6479$). The horizontal dotted line marks the 5% $q$ value cutoff. The vertical dotted line separates the proteins into up- and downregulation. Significantly regulated proteins by $LRRK2^{G2019S}$ in both LCC and Columbia are annotated.

▶

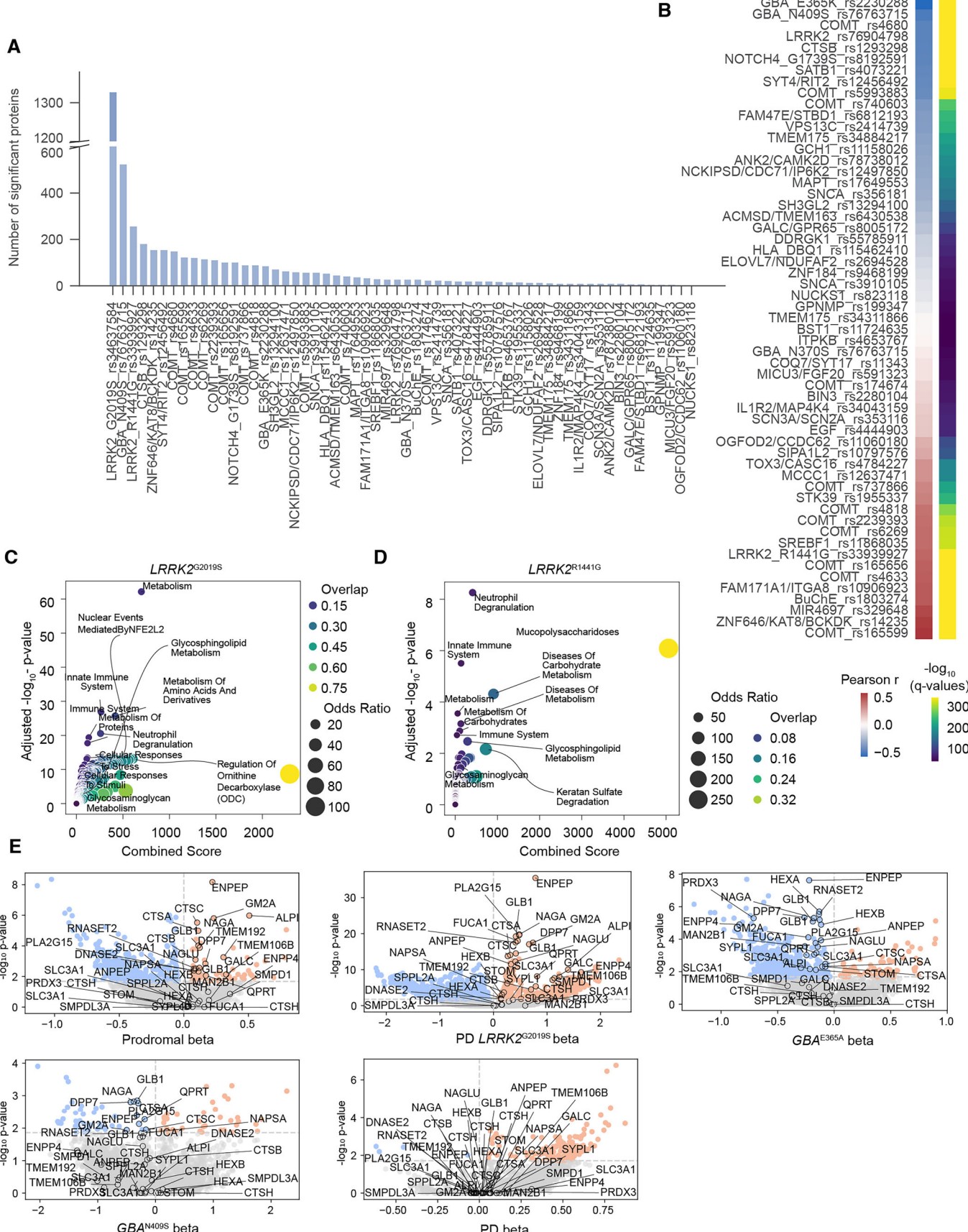

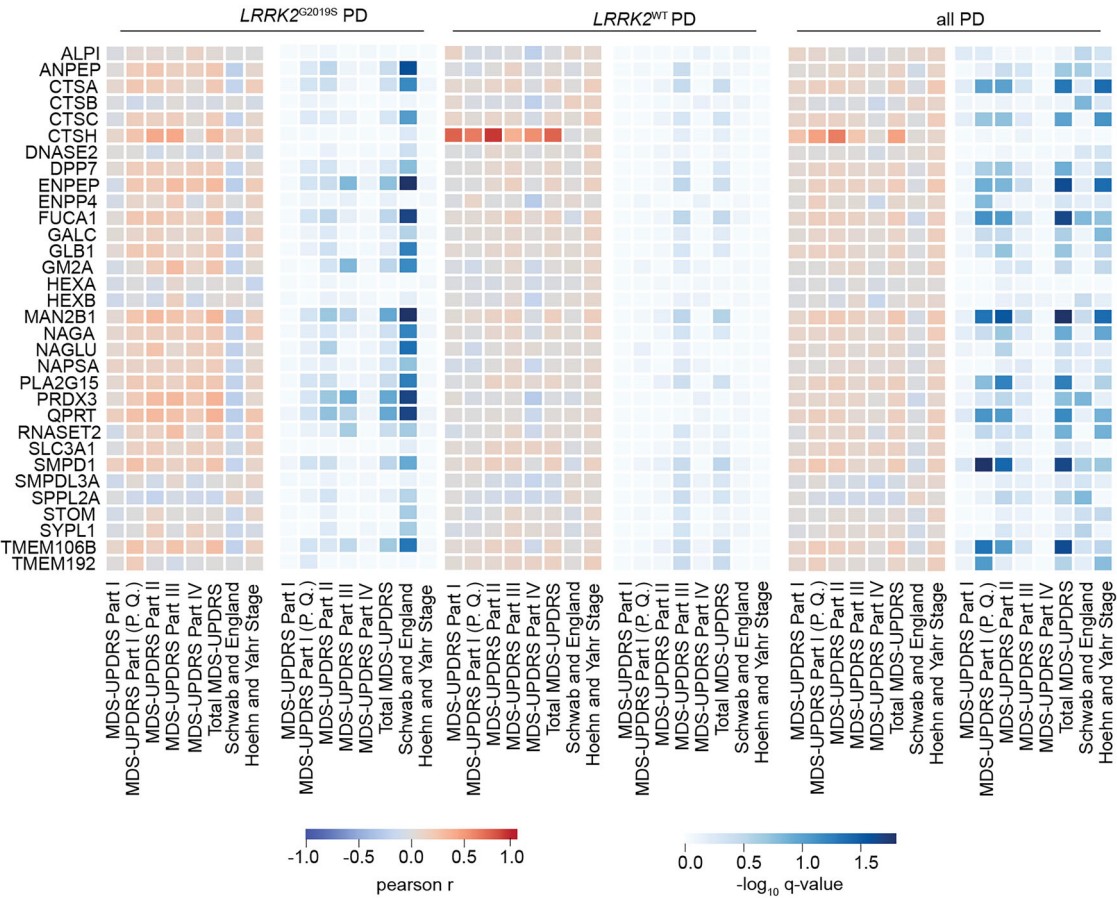

**Figure EV2. Correlations between pathogenic LRRK2-regulated proteins from the LCC and Columbia dataset and clinical disease severity scores.**

Pearson correlation coefficients and $-\log_{10} q$ values for proteins (found significant in the LCC and Columbia cohort, $n = 32$) associated with severity scores in patients with *LRRK2*[G2019S], *LRRK2*[WT] PD and all PD patients.

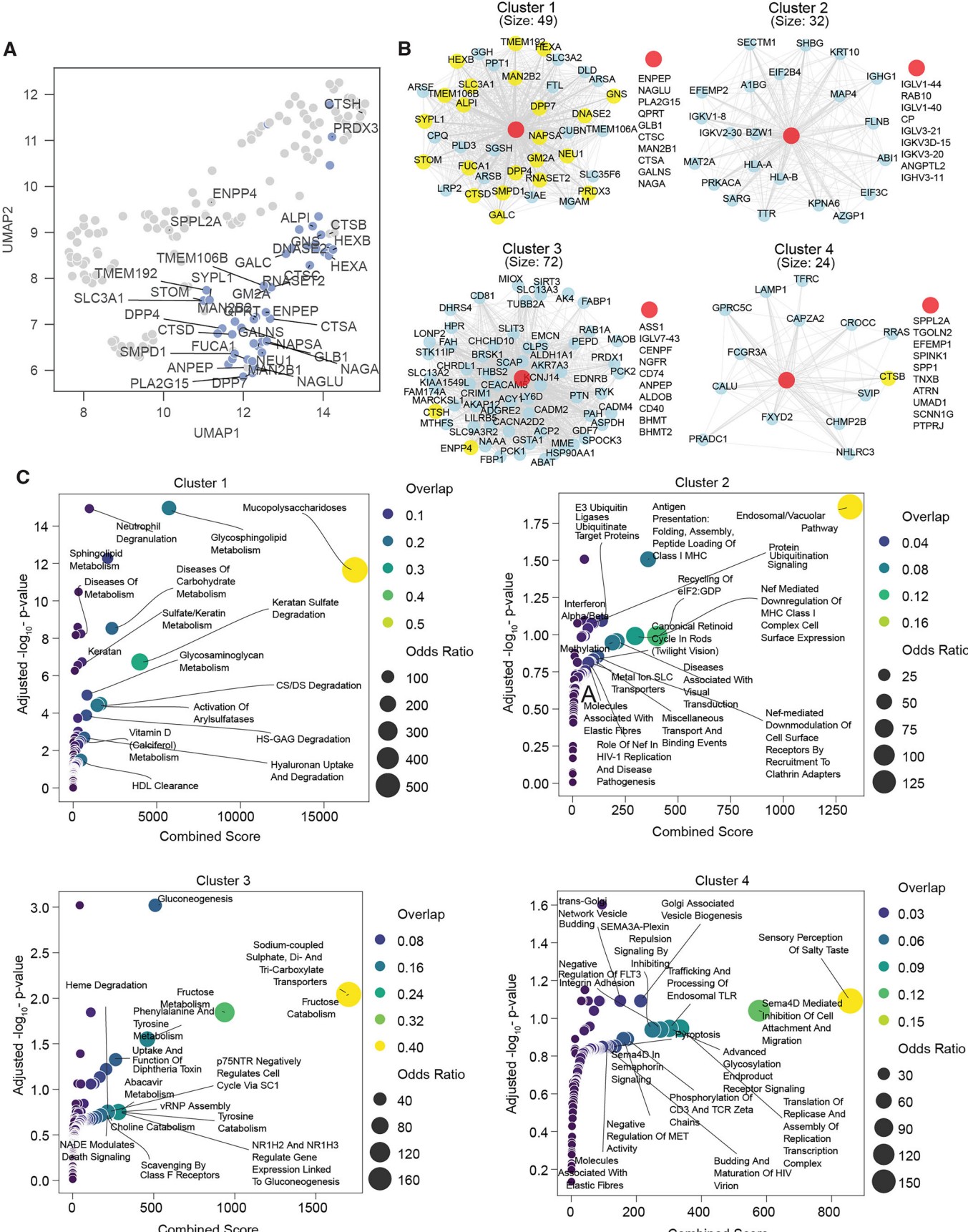

**Figure EV3.   Network analysis of pathogenic LRRK2-regulated proteins.**

(A) UMAP of overlapping pathogenic LRRK2-regulated proteins across three cohorts ($n = 177$), with the significant proteins in LCC and Columbia annotated. (B) Clusters from the network of overlapping pathogenic LRRK2-regulated proteins, with yellow nodes for the significant proteins and red nodes for proteins with the highest degree centrality. Clusters were generated using the Louvain clustering algorithm. (C) Reactome enrichment plots for clusters 1–4, with overlap indicating the proportion of enriched proteins in each term. The most relevant terms were annotated. The overlap illustrates the proportion of enriched proteins in each term. The combined score represents an aggregated metric calculated during enrichment analysis.

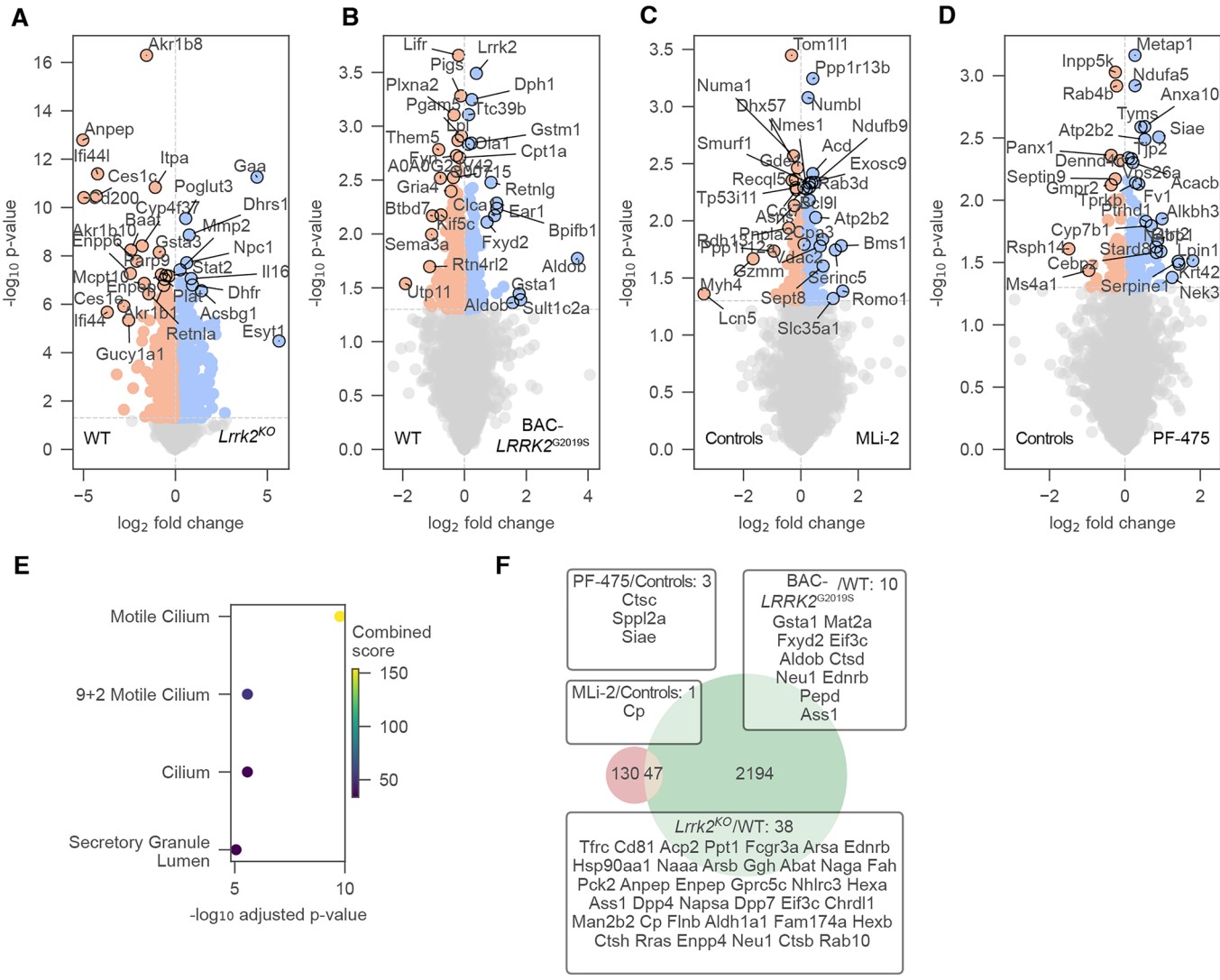

**Figure EV4. Detecting proteome alterations associated with Lrrk2 loss, hyperactivation (G2019S), and kinase inhibition in rat lung.**

(A–D) Volcano plots comparing the lung proteomes of *Lrrk2*^KO vs. *Lrrk2*^WT rats ($n = 8979$) (A), BAC-*LRRK2*^G2019S vs. *Lrrk2*^WT rats ($n = 9187$) (B), MLi-2 vs. Controls ($n = 8447$) (C), and PF-475 vs. Controls ($n = 8447$) (D) with dotted lines indicating 5% $p$ value cutoffs. The vertical dotted line separates the proteins in up- and downregulation. Most relevant proteins are annotated. (E) Enrichment analysis of the GO term "cellular component". Enrichment was performed on significant proteins ($p$ value <5%) in at least two out of four comparisons. The combined score represents an aggregated metric calculated during enrichment analysis. (F) Venn diagram of the union of significant proteins ($p$ value <5%) in *Lrrk2*^KO vs. *Lrrk2*^WT or BAC-*LRRK2*^G2019S vs. *Lrrk2*^WT or MLi-2 vs. Controls or PF-475 vs. Controls (right) and network genes from the human study (left). Intersecting proteins are annotated in a box with their respective comparison.

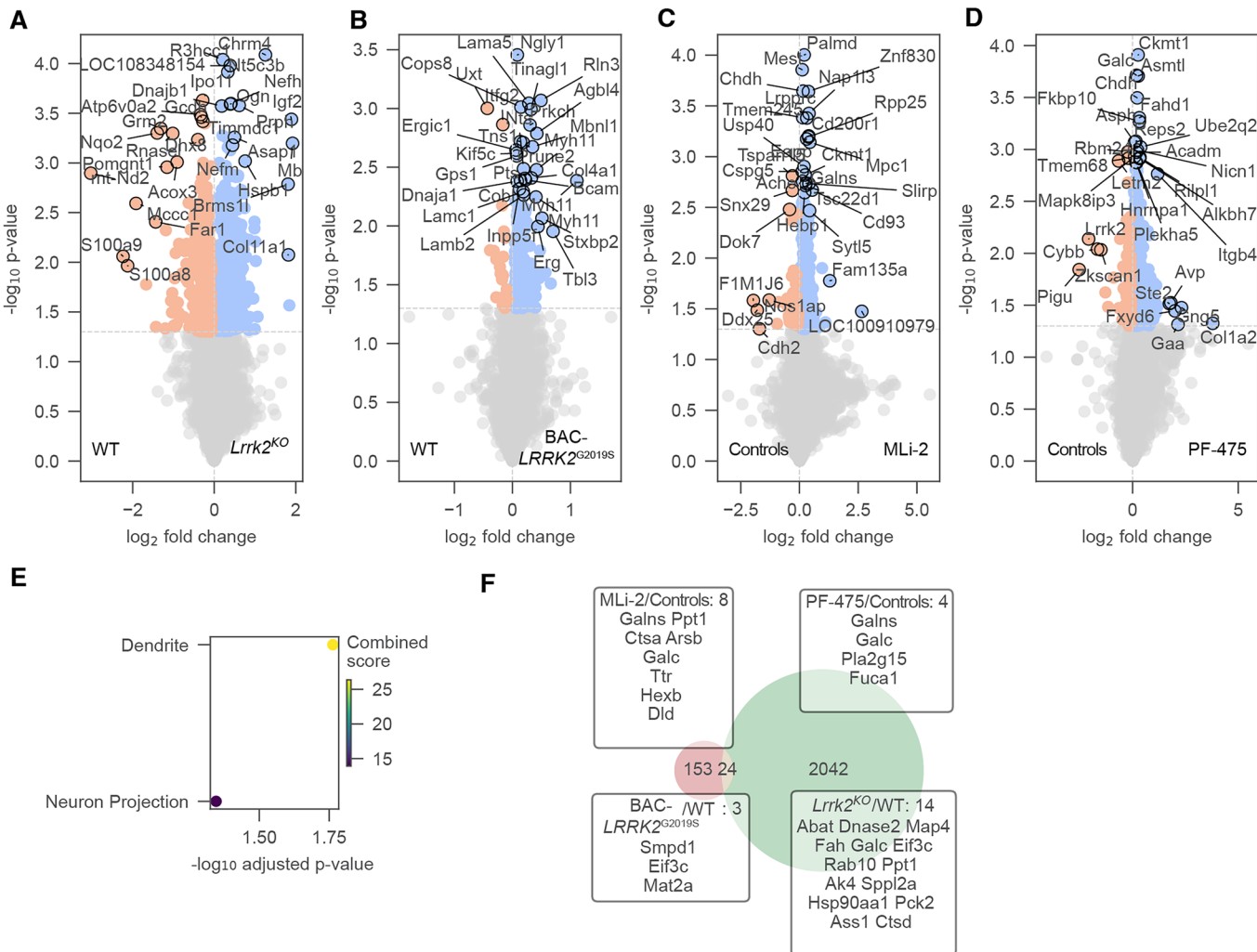

**Figure EV5. Detecting proteome alterations associated with Lrrk2 loss, hyperactivation (G2019S), and kinase inhibition in rat brain.**

(A–D) Volcano plots comparing the brain proteomes of *Lrrk2*^KO vs. *Lrrk2*^WT rats ($n = 8499$) (A) and BAC-*LRRK2*^G2019S vs. *Lrrk2*^WT rats ($n = 8314$) (B), MLi-2 vs. Controls ($n = 8639$) (C), and PF-475 vs. Controls ($n = 8639$) (D) with dotted lines indicating 5% $p$ value cutoffs. The vertical dotted line separates the proteins in up- and down-regulation. Most relevant proteins are annotated. (E) Enrichment analysis of the GO term "cellular component". Enrichment was performed on significant proteins ($p$ value <5%) in at least two comparisons. The combined score represents an aggregated metric calculated during enrichment analysis. (F) Venn diagram of the union of significant proteins ($p$ value <5%) in *Lrrk2*^KO vs. *Lrrk2*^WT or BAC-*LRRK2*^G2019S vs. *Lrrk2*^WT or MLi-2 vs. Controls or PF-475 vs. Controls (right) and network genes from the human study (left). Intersecting proteins are annotated in a box with their respective comparison.

