## [Peer Review File · Molecular Systems Biology]

Multi-cohort, cross-species urinary proteomics reveals LRRK2 dysfunction in Parkinson's disease

Duc Tung Vu, William Sibrán, Andreas Metousis, Laurine Vandewynckel, Basak Eraslan, Liesel Goveas, Ericka Itang, Claire Deldycke, Adriana Figueroa-Garcia, Réginald Lefebvre, Johannes Müller-Reif, Sebastian Virreira-Winter, Marie-Christine Chartier-Harlin, Jean-Marc Taymans, Matthias Mann, and Ozge Karayel

Corresponding author(s): Ozge Karayel (karayel-eren.ozge@gene.com), Jean-Marc Taymans (jean-marc.taymans@univ-lille.fr), Marie-Christine Chartier-Harlin (marie-christine.chartier-harlin@inserm.fr), Matthias Mann (mmann@biochem.mpg.de)

Review Timeline:

Transfer Date:	19th Mar 25
Editorial Decision:	20th Mar 25
Revision Received:	4th Nov 25
Editorial Decision:	2nd Dec 25
Revision Received:	7th Jan 26
Accepted:	13th Jan 26

Editor: Jingyi Hou

Transaction Report: This manuscript was transferred to Molecular Systems Biology following peer review at EMBO Molecular Medicine.

Referee #1 (Remarks for Author):

This is a comprehensive urinary proteomics study looking at readouts of the Parkinson's disease kinase LRRK2. The work seems to have been performed well but some issues are listed below.

The summary incorrectly states that reliable biomarkers for tracking drug engagement or identifying mutation carriers are lacking. Mutation carriers can be sequenced for 100% reliable outcome, no need for expensive proteomic prediction approaches?

Likewise, urinary levels of BMP have been validated as biomarkers for tracking LRRK2 engagement, also levels of LRRK2 ser935 phosphorylation and Rab10. It will be important for this study to show how the models employed improve upon the current state of the art LRRK2 biomarkers, rather than mislead that such things do not exist.

The authors found proteins in known pathways already demonstrated to be regulated by LRRK2. Further discussion is required to highlight any novel aspect of this study. How do results compare to dozens of others similar proteomic studies? The conclusion is that the study has enhanced understanding of the disease, but how? The other conclusion is that the study supports ongoing LRRK2 inhibitor trials, but how? What does this study in urine tell us that we have not learned from CSF and plasma proteomic studies.

The study outcomes are overinterpreted. In the text the authors often refer to candidate biomarkers and biomarker potential, they also routinely state they have found biomarkers of LRRK2 kinase activity, but I don't see if/how LRRK2 kinase activity was measured so this can actually be performed? To me the study seems to identify biomarkers of LRRK2 mutations status, which collectively do not perform as well as gene sequencing, and not biomarkers LRRK2 enzymatic activity. Or the authors have identified potential biomarkers of LRRK2 drug target engagement, but if they perform better or worse than those currently employed in the field is unknown. Either way, how biomarkers relate to LRRK2 kinase activity appears to not be shown.

In the intro it is highlighted that LRRK2 may play a role in both genetic and sporadic PD, but the LRRK2 mutation biomarkers identified did not discriminate sporadic PD, only carriers of LRRK2 mutations? LRRK2 is not actually therefore involved in sporadic PD?

Referee #2 (Comments on Novelty/Model System for Author):

1.High Technical Quality:

The data were acquired using an Evosep One HPLC coupled to a timsTOF Pro 2 mass spectrometer, one of the most advanced LC-MS/MS instruments available. The study employs Data-Independent Acquisition (DIA) mode, ensuring deep proteomic coverage and high data quality. The statistical analyses and machine learning-based classification model are appropriately applied. However, the description of the multiple linear regression model is somewhat vague and should be clarified in the revision.

2.High Novelty:

This study represents the largest urinary proteomic analysis (1,215 samples) of genetic Parkinson's disease in human cohorts. It comprehensively examines a key mutation site in LRRK2, contributing novel insights into PD-associated urinary biomarkers.

3.Medium Medical Impact:

The identified biomarkers from urine samples hold potential for disease diagnosis and pharmacodynamic monitoring but are not applicable for direct therapeutic intervention.

4.Adequate Model System:

Both the human cohort sample size and rat experiments are sufficiently large, ensuring robustness and reliability of the findings.

Referee #2 (Remarks for Author):

In this manuscript, Vu et al. conducted a comprehensive proteomic analysis of urine samples from 1,215 individuals across three cohorts with Parkinson's disease (PD) and/or Leucine-rich repeat kinase 2 (LRRK2) mutations. Using multiple linear regression model, 177 significantly changed proteins were reliably identified. A machine learning model based on 30 selected proteins showed strong classification performance in distinguishing mutation status. Finally, proteomic profiling on urine samples from LRRK2-knockout rats and rats treated with LRRK2 inhibitors validated the ability of biomarker candidates to monitor pharmacodynamic responses.

Overall, the data included in this manuscript is solid. Indeed, this study represents the largest urinary proteomic analysis of genetic PD in human cohorts, with high-quality data. However, the connection between human and rat experiment is weak, and the molecular mechanisms underlying the proteomic findings, particularly in rat experiments, are not well explained. Below are my major and minor aspects:

Major:

1. In the final section of "Results", the authors suggest that "pathogenic LRRK2-regulated

urinary proteins can serve as pharmacodynamic biomarkers". This claim implies that the biomarker candidates identified from human urine cohorts should also exhibit consistent changes in the rat experiments. A key piece of evidence to support this would be a strong correlation between the log₂ fold change of significantly changed proteins in LRRK2G2019S vs. normal in human cohort and the log₂ fold change of mapped rat orthologs in transgenic Lrrk2G2019S vs. WT in rats. However, this correlation is not provided. Simply circling orthologs in rat experiments, as shown in Figure 3C-F, is insufficient, as it only represents protein ID mapping without a quantitative consistency. Therefore, to strengthen the claim, the authors should provide additional analyses demonstrating whether proteomic changes caused by LRRK2 mutation in human cohorts are reproducible in rats.

2. Also in the last section of "Result", the analyses of kidney, lung, and brain tissue proteome from rat experiments are minimal. This is a critical limitation, as a more comprehensive mechanistic understanding of PD and LRRK2 mutations could be derived from examining proteomic changes in these tissues. However, the manuscript lacks in-depth bioinformatic or functional analyses of rat tissue samples to elucidate how LRRK2 mutations or inhibitor treatments affect the tissue proteomes. Additionally, the authors mention "urinary proteome changes were not fully reflected in tissue proteomes" but provided little discussions to support this statement. Therefore, the authors should conduct a deeper interpretation of their rat tissue data, especially in relation to kidney tissue, since the physiological connection between urine and kidney.

Minor:

1. The multiple linear regression model for altered protein discovery is not well-described. The authors mention "We employed MLR to account for confounding factors like age and sex affecting protein intensity. (line 110)" along with some details in "Method" section. However, the full set of variables included in the MLR is unclear. The authors should provide a pseudo-equation to explicitly list all variables, for example:

Protein log₂ intensity ~ age + sex + LRRK2 mutation status + GBA mutation status + PD or not + ...

Additionally, the authors should specify whether different sets of variables were used across different results.

2. The inference "The smaller number of significant proteins for LRRK2R1441G likely reflects the smaller R1441G cohort (26 carriers) compared to the G2019S cohort (305 carriers) in PPMI. (line 114)" lacks solid support. The authors should firstly consider whether these two mutations have different biological effects. It is possible that LRRK2R1441G exerts a milder influence on LRRK2 function, whereas LRRK2G2019S has a more pronounced effect. The authors provide evidence that these two mutations impact LRRK2 similarly before attributing the difference solely to the sample sizes.

3. Some methodological terms used in the "Results" section are inconsistent with those in

the "Method" section, which may cause confusion. For example:

- "Regularized linear regression" in line 160 should explicitly state "Ridge regression" as mentioned in the "Method" section.
- "K-fold cross validation" in line 205 should match the more detailed description in the "Method" section: "a repeated stratified fold with 5 splits and 3 repeats".

4. In line 217, the authors stated, "The same ML strategy used to differentiate between PD patients and controls, as well as genetic PD and non-manifesting carriers, performed poorly, only slightly better than random chance." However, they do not provide any evidence to support this claim. The authors should include relevant performance metric ROC curves and AUC values as supplementary materials.

5. The authors should provide some information about the mechanisms of action of Lrrk2 inhibitors, which could be helpful to explain the results comparing to Lrrk2KO.

6. For correlation plots like Figure 3C-F, the author should provide the global correlation for each comparison. For example, in Figure 3C, the Pearson correlation for significantly changed proteins is -0.73. If the correlation of all proteins is near 0, then -0.73 represents a reasonable strong negative correlation. However, if the global correlation is already around -0.6, -0.73 may not be as meaningful in demonstrating a strong negative correlation.

7. There are less description on MLI-2 results. The authors should consider adding two supplementary correlation figures: Lrrk2KO vs. MLI-2; Lrrk2G2019S vs. MLI-2.

8. The caption of Figure EV4 "Peripheral effects of Lrrk2 deficiency, hyperactivation, and inhibition in rat kidney, lung, and brain" is inaccurate, since Figure EV4A and 4B are the plots for urine samples.

9. The color scheme of kidney, lung, and brain proteins in Figure 3 H-K is confusing. Initially, it appears that many kidney proteins are shown in blue (suggesting down-regulation) despite having a positive log₂ fold change, or in red (suggesting up-regulation) despite having a negative log₂ fold change. Upon closer look, it becomes clear they are significantly changed proteins found in urine samples then mapped to other tissues. To improve clarity, the authors should refine the color labels to make them more specific (e.g., "downregulated in urine samples"), or consider alternative visualization methods, such as heatmaps, to better illustrate the data.

10. Authors should carefully proofread the manuscript to correct minor typographical and formatting errors. Examples include: "LRRK2 inhibitors" in "Summary" should be "Lrrk2 inhibitors"; "2" in "log₂" should be subscripted in the main text and figures.

Referee #3 (Remarks for Author):

The manuscript presents a well-executed study integrating multi-cohort urinary proteomics with machine learning to identify biomarkers associated with LRRK2 kinase activity in Parkinson's disease (PD). The authors provide a comprehensive urine proteome analysis, demonstrating that a panel of 30 urinary proteins linked to lysosomal dysfunction and glycosphingolipid metabolism can classify LRRK2 mutation carriers with high accuracy. Additionally, validation in a rat model supports the utility of this biomarker panel in assessing pharmacodynamic responses to LRRK2 inhibition. This work demonstrated a nice study model to offers both novel molecular insights into PD pathogenesis and direct implications for companion biomarker-driven clinical applications in monitoring the treatment efficacy using noninvasive specimens. On the technical aspect, the work demonstrated deep and large-scale urine proteome (>3000 proteins per individuals) by state-of-art DIA approach, and its integration with machine learning and translational validation. The manuscript presents a strong study with clear translational potential, but addressing the following comments will improve its clinical relevance and clarity on methodological robustness before publication:.

1. This study includes multiple cohorts which may encountered variations in sample collection, storage, processing and environmental factors, particularly when the sizes of the three cohorts are quite difference. Was there any batch effect correction performed for the proteome datasets? Given that the batch effect is a common factor for clinical proteomics spanning multiple cohorts, detailed elaboration on the batch correction processing and additional discussion on potential batch effects, sample processing variability, and cross-cohort reproducibility would be beneficial. The Authors may need to add the information in Supporting Information.
2. The study successfully identifies LRRK2-associated urinary biomarkers. How would these biomarkers compare to existing PD diagnostic/prognostic markers in cerebrospinal fluid or blood? How do these urinary biomarkers compare with other established LRRK2 activity biomarkers (e.g., phosphorylated Rab proteins in peripheral blood cells)?
3. Given that urine reflects kidney function and systemic metabolic processes, there are potential confounders and urinary biomarkers may not have high specificity to reflect LRRK2 activity in neurons rather than renal pathology or systemic effects. Were there any correlations between biomarker levels and renal function markers in the dataset? Additionally, since urine contains proteins filtered from multiple tissues, a more explicit discussion on whether these changes specifically reflect neuronal LRRK2 activity versus systemic alterations is needed.

Minor points:

1. For this large-scale study, quality controls for sample collection and data generation in LC-MS/MS workflow are critical. Adding information on quality control measures used will be helpful to ensure reproducibility.
2. Figure 1 (Volcano plots) could benefit from highlighting key proteins rather than showing all 177 proteins.
3. Figure 3 (panel C-F). The X- and Y-axis titles can be further revised for better clarity.

20th Mar 2025

Manuscript Number: MSB-2025-12976-T

Title: Urinary proteome profiling in rats and humans unveils biomarkers of LRRK2 kinase activity in Parkinson's disease

Author: Duc Tung Vu

Marie-Christine Chartier-Harlin

William Sibrán

Andreas Metousis

Laurine Vandewynckel

Basak Eraslan

Liesel Goveas

Ericka Itang

Claire Deldycke

Adriana Figueroa-Garcia

Reginald Lefebvre

Johannes Müller-Reif

Sebastian Virreira-Winter

Jean-Marc Taymans

Matthias Mann

Ozge Karayel

Dear Dr. Karayel,

Thank you for submitting your work to Molecular Systems Biology. Based on the peer-review reports from EMBO Mol Med, we would like to invite you to submit a revised version of your manuscript for further consideration at our journal.

Reviewers #2 and #3, both from the proteomics field, are generally supportive of your work. However, Reviewer #1, who specializes in the PD field, acknowledges the technical quality of the study but raises concerns regarding its direct medical impact. Considering the balance of the reviews and the scope of Molecular Systems Biology, we think we can invite you to submit a major revision to address the concerns raised.

In particular, we would ask you to highlight the novelty and medical relevance of your study more clearly and ensure it is properly contextualized within the existing literature. Additionally, Reviewer #2's concerns related to the rat experiment analyses should be carefully addressed. All the other issues regarding technical aspects, presentation, and potential overstatements must be thoroughly resolved.

As you may already know, our editorial policy allows in principle a single round of major revision, and it is therefore essential to provide responses to the reviewers' comments that are as complete as possible. Please feel free to contact me in case you would like to discuss in further detail any of the issues raised by the reviewers.

On a more editorial level, we would ask you to address the following issues:

- Please provide a .docx formatted version of the manuscript text (including legends for main figures, EV figures and tables). Please make sure that the changes are highlighted to be clearly visible.

- Please provide individual production quality figure files as .eps, .tif, .jpg (one file per figure).

- Please provide a .docx formatted letter INCLUDING the reviewers' reports and your detailed point-by-point responses to their comments. As part of the EMBO Press transparent editorial process, the point-by-point response is part of the Review Process File (RPF), which will be published alongside your paper.

- Please note that all corresponding authors are required to supply an ORCID ID for their name upon submission of a revised manuscript.

- We replaced Supplementary Information with Expanded View (EV) Figures and Tables that are collapsible/expandable online (see examples in <http://msb.embopress.org/content/11/6/812>). A maximum of 5 EV Figures can be typeset. EV Figures should be cited as 'Figure EV1, Figure EV2' etc... in the text and their respective legends should be included in the main text after the legends of regular figures.

Additional Tables/Datasets should be labeled and referred to as Table EV1, Dataset EV1, etc. Legends have to be provided in a separate tab in case of .xls files. Alternatively, the legend can be supplied as a separate text file (README) and zipped together

with the Table/Dataset file.

For the figures and tables that you do NOT wish to display as Expanded View figures, they should be bundled together with their legends in a single PDF file called *Appendix*, which should start with a short Table of Content. Each legend should be below the corresponding Figure/Table in the Appendix. Appendix figures and tables should be referred to in the main text as: "Appendix Figure S1, Appendix Figure S2, Appendix Table S1" etc. See detailed instructions regarding expanded view here: <https://www.embopress.org/page/journal/17444292/authorguide#expandedview>.

-Before submitting your revision, primary datasets (and computer code, where appropriate) produced in this study need to be deposited in an appropriate public database (see [http://msb.embopress.org/authorguide - dataavailability](http://msb.embopress.org/authorguide-dataavailability)

<https://www.embopress.org/page/journal/17444292/authorguide#dataavailability>).

The accession numbers and database should be listed in a formal "Data Availability" section (placed after Materials & Method) that follows the model below (see also <https://www.embopress.org/page/journal/17444292/authorguide#dataavailability>). Please note that the Data Availability Section is restricted to new primary data that are part of this study.

Data availability

- RNA-Seq data: Gene Expression Omnibus GSE46843 (<https://www.ncbi.nlm.nih.gov/geo/query/acc.cgi?acc=GSE46843>)

- [data type]: [name of the resource] [accession number/identifier/doi] ([URL or identifiers.org/DATABASE:ACCESSION])

-At EMBO Press we ask authors to provide source data for the main figures. Our source data coordinator will contact you to discuss which figure panels we would need source data for and will also provide you with helpful tips on how to upload and organize the files.

- Our journal encourages inclusion of *data citations in the reference list* to directly cite datasets that were re-used and obtained from public databases. Data citations in the article text are distinct from normal bibliographical citations and should directly link to the database records from which the data can be accessed. In the main text, data citations are formatted as follows: "Data ref: Smith et al, 2001". In the Reference list, data citations must be labeled with "[DATASET]". A data reference must provide the database name, accession number/identifiers and a resolvable link to the landing page from which the data can be accessed at the end of the reference. Further instructions are available at .

- We updated our journal's competing interests policy in January 2022 and request authors to consider both actual and perceived competing interests. Please review the policy <https://www.embopress.org/competing-interests> and update your competing interests if necessary.

Please use the heading "Disclosure statement and competing interests".

- All Materials and Methods need to be described in the main text using our 'Structured Methods' format. According to this format, the Methods section includes a Reagents and Tools Table (listing key reagents, experimental models, software and relevant equipment and including their sources and relevant identifiers) followed by a Methods and Protocols section describing the methods, ideally using a step-by-step protocol format. The aim is to facilitate adoption of the methodologies across labs. Please download and fill our Reagents and Tools Table template (.docx), which you can find in our author guidelines: <https://www.embopress.org/page/journal/17444292/authorguide#structuredmethods>.

-Regarding data quantification:

Please ensure to specify the name of the statistical test used to generate error bars and P values, the number (n) of independent experiments (please specify technical or biological replicates) underlying each data point and the test used to calculate p-values in each figure legend. Discussion of statistical methodology can be reported in the materials and methods section, but figure legends should contain a basic description of n, P and the test applied.

Graphs must include a description of the bars and the error bars (s.d., s.e.m.).

- Please provide a "standfirst text" summarizing the study in one or two sentences (approximately 250 characters, including space), three to four "bullet points" highlighting the main findings and a "synopsis image" (550px width and 400-600 px height, PNG format) to highlight the paper on our homepage.

Here are a couple of examples:

<https://www.embopress.org/doi/10.15252/msb.20199356>

<https://www.embopress.org/doi/10.15252/msb.20209475>

<https://www.embopress.org/doi/10.15252/msb.209495>

When you resubmit your manuscript, please download our CHECKLIST (<https://www.embopress.org/pb-assets/embo-site/EMBO%20Press%20Author%20Checklist-1642513524327.xlsx>) and include the completed form in your submission.

Please note that the Author Checklist will be published alongside the paper as part of the transparent process (<https://www.embopress.org/page/journal/17444292/authorguide#transparentprocess>).

If you feel you can satisfactorily deal with these points and those listed by the referees, you may wish to submit a revised version of your manuscript. Please attach a covering letter giving details of the way in which you have handled each of the points raised by the referees. A revised manuscript will be once again subject to review and you probably understand that we can give you no guarantee at this stage that the eventual outcome will be favorable.

I look forward to receiving your revised manuscript soon.

Sincerely,
Jingyi

Jingyi Hou, PhD
Senior Editor
Molecular Systems Biology

We realize that it is difficult to revise to a specific deadline. In the interest of protecting the conceptual advance provided by the work, we recommend a revision within 3 months (18th Jun 2025). Please discuss the revision progress ahead of this time with the editor if you require more time to complete the revisions.

IMPORTANT: When you send your revision, we will require the following items:

1. the manuscript text in LaTeX, RTF or MS Word format
2. a letter with a detailed description of the changes made in response to the referees. Please specify clearly the exact places in the text (pages and paragraphs) where each change has been made in response to each specific comment given
3. three to four 'bullet points' highlighting the main findings of your study
4. a short 'blurb' text summarizing in two sentences the study (max. 250 characters)
5. a 'thumbnail image' (550px width and max 400px height, Illustrator, PowerPoint or jpeg format), which can be used as 'visual title' for the synopsis section of your paper.
6. Please include an author contributions statement after the Acknowledgements section (see <https://www.embopress.org/page/journal/17444292/authorguide>)
7. Please complete the CHECKLIST available at (<https://bit.ly/EMBOPressAuthorChecklist>). Please note that the Author Checklist will be published alongside the paper as part of the transparent process (<https://www.embopress.org/page/journal/17444292/authorguide#transparentprocess>).
8. When assembling figures, please refer to our figure preparation guideline in order to ensure proper formatting and readability in print as well as on screen: <https://bit.ly/EMBOPressFigurePreparationGuideline>
See also figure legend guidelines: <https://www.embopress.org/page/journal/17444292/authorguide#figureformat>
9. Please note that corresponding authors are required to supply an ORCID ID for their name upon submission of a revised manuscript (EMBO Press signed a joint statement to encourage ORCID adoption). (<https://www.embopress.org/page/journal/17444292/authorguide#editorialprocess>)
Currently, our records indicate that there is no ORCID associated with your account.

Please click the link below to provide an ORCID:
Link Not Available

11. Include a Reagents and Tools Table as part of the Methods section, which can be downloaded from our author guidelines (<https://www.embopress.org/page/journal/17444292/authorguide#structuredmethods>)

*** PLEASE NOTE *** As part of the EMBO Press transparent editorial process initiative (see our Editorial at <https://dx.doi.org/10.1038/msb.2010.72>), Molecular Systems Biology publishes online a Review Process File with each accepted

manuscripts. This file will be published in conjunction with your paper and will include the anonymous referee reports, your point-by-point response and all pertinent correspondence relating to the manuscript. If you do NOT want this File to be published, please inform the editorial office at msb@embo.org within 14 days upon receipt of the present letter.

November 3, 2025

Response to Reviewers' Comments: *Multi-Cohort, Cross-Species Urinary Proteomics Reveals Signatures of LRRK2 Dysfunction in Parkinson's Disease*

We thank all reviewers for their insightful and constructive comments. In response, over the last eight months, we have revised and restructured our manuscript, adding new analyses, clarifying methods, and expanding the discussion. We trust that they demonstrate our commitment to addressing the concerns raised with rigor and transparency and hope that the reviewers and editors will find them satisfactory.

Executive summary:

The reviewers asked us to: (i) clarify how our urinary protein markers relate to established LRRK2 activity readouts and to avoid implying they replace genetic testing; (ii) rule out batch effects and renal pathology as confounders; (iii) quantitatively validate cross-species reproducibility and expand on rat tissue proteomics analyses; (iv) establish novelty versus existing biofluid proteomics; and (v) deepen the biological interpretation (tissue origins; systemic vs. neuronal LRRK2 dysfunction). We have comprehensively addressed these concerns as follows:

- Clarified scope and positioning: We have revised language throughout to clarify that our biomarker candidates reflect downstream consequences of LRRK2 dysfunction, not direct kinase activity. Sequencing remains the standard for identifying mutation carriers; our approach is complementary for pathway-centric stratification and pharmacodynamic monitoring. We discuss our findings in the context of established markers (urinary BMPs and pRab10/pLRRK2) and add a targeted MS analysis in the same urine cohort that confirms higher total RAB10/LRRK2, whereas our phosphopeptide quantification was less robust in urine than in cells.
- Expanded cross-biofluid validation: Re-analysis of PPMI plasma/CSF (DIA-MS and Olink) reveals that the overlap with urine is limited, reflecting both the shallower DIA-MS coverage in plasma (186 proteins) and CSF (291 proteins) and the distinct biology captured in urine. Most LRRK2-dependent changes and 177 network proteins identified in urine were not detected in these biofluids, with only two proteins shared across all matrices. This highlights urine as a complementary source providing access to LRRK2-related pathways not readily measurable in plasma or CSF.
- Quantified cross-species reproducibility: We provide fold-change correlations between rat and human G2019S (Pearson r 0.26), identify a conserved lysosomal signature (e.g., cathepsins, glycosphingolipid enzymes) elevated in both species and show reversal with LRRK2 inhibition in rat urine. A classifier trained on rat-derived features generalizes to humans (ROC AUC 0.75), providing evidence of translational validity of our findings and for PD monitoring.
- Deepened rat analyses and tissue specificity: In response to requests for more granular rat data, we reanalyzed tissue-resolved proteomics. We show that the urinary signature aligns with tissue changes (enrichment of lysosomal/vesicular trafficking pathways), that directionality is consistent across matrices. Pharmacological LRRK2 inhibition attenuates

these shifts. These results provide context to urinary readouts and strengthen the link between peripheral and central LRRK2 pathway dysfunction.

- Strengthened batch control and methods transparency. Primary statistics were performed within the cohort; ML feature discovery was repeated with/without pyComBat batch correction and remained stable across instruments and protocols. We add the explicit MLR equation, precise CV scheme, QC/outlier handling, pooled-control variabilities, and added details of ML-based classification of PD vs. control.
- Tissue-of-origin analysis indicates that urinary proteins reflect both renal and extra-renal sources, including brain-relevant components. Several LRRK2-regulated proteins identified in urine show concordant regulation in rat brain and lung, supporting a systemic signature rather than a kidney-specific effect. Correlations between these urinary markers and renal function indicators (albumin, cystatin C, UMOD) are weak and not genotype-dependent, arguing against renal pathology as the main source of variation.
- Added clinical insights: While the carrier classifier achieves mean ROC AUC 0.91 (88% sensitivity; 80% specificity) across three cohorts, “false positives” in PPMI often carry other LRRK2 or GBA variants (23/30), and some idiopathic PD/prodromal cases share the G2019S-like signature—consistent with broader LRRK2-pathway involvement beyond G2019S.
- These revisions (highlighted in red in the manuscript) preserve our central conclusions while substantially strengthening statistical rigor, biological interpretation, and clinical relevance. We have added multiple new analyses (Figure 1, 3, 4 and 5, Figure EV 1, 4, and 5, Revision Figures 1-5, Appendix Figures S1, S7, S8, S10, S12, S13, S14 and S15) and expanded our methods and discussion section for limitations and transparency.

In direct response regarding the novelty and utility of our study:

- Largest urinary proteomics resource in PD: In-depth profiling from 1,215 participants and three independent cohorts, constituting (to our knowledge) the largest PD urinary proteomics dataset to date, supported by a robust computational framework.
- Genetics-to-proteome mapping: Integration of WGS from >500 participants to map how 58 risk variants across 45 loci shape the urinary proteome—moving beyond single-variant analyses. LRRK2—especially G2019S—exerts the strongest effects, with inverse patterns for common GBA variants.
- Systems-level LRRK2 biology: A 177-protein network capturing LRRK2-linked lysosomal, glycosphingolipid, immune, and trafficking pathways—extending interpretation beyond single-analyte markers (BMPs, pRab10) distilled to a 30-protein, cohort-agnostic panel
- Cohort-agnostic biomarker panel: ML distilled a 30-protein urinary panel that generalizes across human cohorts, classify G2019S carriers (mean AUC 0.91) and identify “G2019S-like” profiles beyond genotype.
- Cross-species translation: Concordant signatures in *in vivo* rat models (KO, BAC-G2019S) and after kinase inhibition (MLi-2, PF-475) provide evidence that pathway changes are LRRK2-regulated. Matched tissue proteomics (kidney, lung, brain) reveal organ-level sources for urinary signals and demonstrate overlap with human urine (strongest in kidney), including brain-relevant proteins.

- Pharmacodynamic utility: Rat-derived perturbations predict human LRRK2 mutation status (AUC 0.75) and reverse with LRRK2 inhibition, supporting a non-invasive PD pharmacodynamic readout.
- Clinical complementarity: Urine captures brain-relevant and kidney-proximal pathway states, yielding non-invasive biomarker candidates that complement BMPs and pRab10 for target engagement and may enable pathway-centric stratification and monitoring in LRRK2-directed trials.

Point-by-point responses:

Reviewer 1

Major remarks

1. *The summary incorrectly states that reliable biomarkers for tracking drug engagement or identifying mutation carriers are lacking. Mutation carriers can be sequenced for 100% reliable outcome, no need for expensive proteomic prediction approaches?*

We agree that sequencing provides a reliable means of identifying mutation carriers and our study in no way aims to replace sequencing. Instead, unbiased urinary proteomics offers complementary clinical advantages, independent of mutation status, in two key areas: (1) stratifying individuals with LRRK2 dysfunction but without a *LRRK2* mutation (either idiopathic or carrying a different risk variant) for treatments targeting LRRK2 activity and related pathways, and (2) serving as pharmacodynamic markers to help monitor the effectiveness of such treatments. We have included the following changes in lines 270-285:

Our machine learning model excelled in discriminating LRRK2 mutation carriers from non-carriers, achieving a mean ROC AUC of 0.91, 88% sensitivity, and 80% specificity across three cohorts, with 30 false positives when applied to the PPMI cohort (Figure 2). Those false positives were classified by the model as LRRK2 G2019S-like based on their urinary proteomic signatures even without the G2019S mutation. Validating our approach, 23 false positives carried either a different LRRK2 mutation (R1441G and N2081D) or a GBA mutation (N409S or T408M) or both genes (Appendix Figure S7). Interestingly, the N2081D mutation in the LRRK2 gene is also associated with increased kinase activity, similar to G2019S. Specifically, N2081D disrupts the interaction between the LRRK2 protein's Leucine-rich repeat (LRR) domain and the kinase domain, exposing the active site and enhancing the kinase's ability to phosphorylate its substrates, including RAB proteins as we have recently shown (Heaton et al., 2025). This mutation is linked to an increased risk of inflammatory bowel disease (IBD) and, while also associated with PD, the association with IBD is stronger (Hui et al., 2018), suggesting LRRK2 pathway dysregulation in diseases beyond PD is measurable in urine PD.

2. *Likewise, urinary levels of BMP have been validated as biomarkers for tracking LRRK2 engagement, also levels of LRRK2 ser935 phosphorylation and Rab10. It will be important for this study to show how the models employed improve upon the current state of the art LRRK2 biomarkers, rather than mislead that such things do not exist.*

We thank the reviewer for this important comment. We now explicitly acknowledge existing *LRRK2* biomarkers—bis(monoacylglycerol)phosphates (BMPs) and RAB10/*LRRK2* phosphorylations—which serve as established readouts for *LRRK2* target engagement in both preclinical and clinical studies. We have added in lines 499-507:

Existing LRRK2-pathway biomarkers—phosphorylation readouts (RAB10 Thr73 as well as LRRK2 Ser935 and Ser1292) and urinary BMPs (notably di-22:6-BMP) - are proven pharmacodynamic tools for confirming target engagement and guiding dosing in trials (Alcalay et al., 2020; Gomes et al., 2023; Ö. Karayel et al., 2020; Taymans, Mutez, et al., 2023). However, each has limits: urinary BMPs reflect peripheral lysosomal lipid handling and are only consistently elevated in genotypes that hyperactivate the LRRK2 pathway (e.g., LRRK2/VPS35 axis) and pRAB10 and pLRRK2 can be difficult to quantify in biofluids, especially in urine, without enrichment. LRRK2 Ser1292 is particularly challenging to detect by mass spectrometry as trypsinization yields a very short peptide (Rideout et al., 2020).

To clarify how our approach complements these biomarkers we did the following analysis lines 507-516:

We analyzed our unpublished targeted mass spectrometry data from the same PPMI urine cohort, quantifying LRRK2-phosphorylated RAB10 (Thr73) and LRRK2 phosphorylation sites (Ser910, Ser935, Ser973) (Appendix Figure S14). This showed elevated levels of both unmodified and phosphorylated RAB10 and LRRK2 peptides in G2019S carriers, with increases primarily at the total protein level. However, accurate quantification of these low-abundance phosphopeptides was challenging due to weak signals difficult to distinguish from background noise. Our findings corroborate our urinary proteomics data from PPMI showing significantly elevated RAB10 and LRRK2 proteins in G2019S carriers (EV3B; Dataset EV1), consistent with recent observations by Taymans et al., 2023. These results suggest that, unlike in peripheral blood cells where phosphorylated RAB10 and LRRK2 are robust biomarkers (Jennings et al., 2022; Ö. Karayel et al., 2020), their phosphorylated forms may not be ideal for detecting LRRK2 mutation carriers in urine.

Our study employs an unbiased, systems-level proteomic approach that complements rather than replaces existing biomarkers. While urinary BMPs are well-established pharmacodynamic biomarkers for *LRRK2* inhibitor monitoring, they have limited utility for distinguishing among PD genotypes, hence pathway-centric patient stratification. Our urinary protein panel provides (changes in lines 516-525):

(1) broader mechanistic insights into LRRK2 dysfunction (immune signaling, vesicle trafficking, mitochondrial function, protein degradation), (2) genotype-informed patient stratification capabilities, and (3) identification of more abundant, robustly quantifiable proteins suitable for clinical translation via targeted immunoassays (Appendix Figure S14C). Together, these biomarkers form a complementary, tiered strategy (Appendix Figure S7) for both mechanistic understanding and clinical application.

- 3. The authors found proteins in known pathways already demonstrated to be regulated by LRRK2. Further discussion is required to highlight any novel aspect of this study. How do results compare to dozens of others similar proteomic studies? The conclusion is that the study has enhanced understanding of the disease, but how? The other conclusion is that the*

study supports ongoing LRRK2 inhibitor trials, but how? What does this study in urine tell us that we have not learned from CSF and plasma proteomic studies.

We thank the reviewer for raising these important questions and addressed them as follows (executive summary and revised manuscript lines 156-167):

To demonstrate the novelty and breadth of our study, we have now included a systems biology approach that integrates whole genome sequencing (WGS) data from over 500 PPMI participants (Figure 1B-C, EV1A-B, Dataset EV1). This included 58 distinct mutations across 45 genes beyond LRRK2 and GBA—such as COMT, which encodes catechol-O-methyltransferase, a key regulator of dopamine metabolism. This systematic assessment revealed that LRRK2 mutations, particularly G2019S, had the most pronounced effect on the urinary proteome (Figure EV1A), and uncovered both convergent and divergent proteomic patterns between LRRK2 and other genetic variants, offering novel insights into shared and distinct molecular mechanisms.

To further address the reviewer's question regarding novel insights from our urinary proteomics study compared to CSF and plasma studies- as well as a related comment from Reviewer 3 -we have now also reanalyzed DIA-MS (Athieniti et al., 2025) and Olink datasets from the PPMI cohort (Rutledge et al., 2024). The DIA-MS analysis showed limited proteome depth - only 186 proteins in plasma and 291 in CSF (Revision Figure 1A), with minimal overlap with urine (2.91% and 4.5%, respectively; Revision Figure 1B). This identified only 4 and 13 differentially abundant proteins in plasma and CSF of G2019S carriers, respectively (Revision Figure 1C). Notably, cathepsins CTSB and CTSD showed consistent regulation in both urine and CSF, aligning with our previous findings (O. Karayel et al., 2022). For Olink data, while 17 and 56 proteins were significantly regulated in plasma and CSF respectively, none of the 177 network genes consistently regulated across our three urinary cohorts were included in Olink studies (Revision Figure 1D-E). (Rutledge et al., 2024) identified only two proteins—VEGFA and SERPINA11—as consistently regulated across multiple biofluids. Limited cross-biofluid overlap likely reflects distinct protein compositions and the sensitivities of these platforms (changes in lines 169-170):

Given LRRK2's high renal expression, urine offers a unique, non-invasive matrix to interrogate pathogenic LRRK2 alterations in depth— borne out by our finding that LRRK2 mutations showed the strongest urinary effects among the 58 variants evaluated (Figure 1B).

Thus, the score and depth of our urinary proteomics study deliver insights not captured by previous biofluid-based studies and validates and extends earlier clinical proteomics work in PD. To our knowledge, this is the largest urinary proteomics dataset analyzed to date for a neurodegenerative disease, and is supported by a robust computational framework. This establishes a strong foundation for future large-scale urinary proteomics research and opens new opportunities for applying this approach well beyond renal pathology.

Revision Figure 1. Analysis of DIA-MS and Olink CSF and plasma datasets from the PPMI cohort.

A) Number of identified proteins in DIA-MS CSF and plasma datasets.

B) Intersection of quantified proteins across different biofluids using DIA-MS.

C) Volcano plots illustrating beta values from linear regression analysis versus $-\log_{10}$ p-values for comparing $LRRK2^{WT}$ with $LRRK2^{G2019S}$ using the DIA-MS dataset. Significant proteins (q -value $<5\%$) are highlighted and annotated.

D) Same as B) but for Olink dataset.

E) Same as C) but for Olink dataset.

4. *The study outcomes are overinterpreted. In the text the authors often refer to candidate biomarkers and biomarker potential, they also routinely state they have found biomarkers of LRRK2 kinase activity, but I don't see if/how LRRK2 kinase activity was measured so this can actually be performed? To me the study seems to identify biomarkers of LRRK2 mutation status, which collectively do not perform as well as gene sequencing, and not biomarkers of LRRK2 enzymatic activity. Or the authors have identified potential biomarkers of LRRK2 drug target engagement, but if they perform better or worse than those currently employed in the field is unknown. Either way, how biomarkers relate to LRRK2 kinase activity appears to not be shown.*

We appreciate and agree with the reviewer's concern regarding the interpretation of our results as biomarkers of LRRK2 kinase activity. To avoid overinterpretation, we revised the text throughout to clarify that the alterations we detect are associated with pathogenic *LRRK2* mutations and are therefore indicative of downstream changes related to pathogenic variants and LRRK2 dysfunction, rather than direct measures of enzymatic activity as mentioned above.

Furthermore, to investigate whether our proteomic signatures might serve as pharmacodynamic markers, we conducted a rat urine study using two potent, brain-penetrant LRRK2 inhibitors (MLi-2 and PF-475), since we lack access to human samples from clinical trials (changes in lines 355-358):

Consistent with previous findings by our group (Virreira Winter et al., 2021) and others (Hadisurya et al., 2023), we observed elevated levels of lysosomal proteins and glycosphingolipid metabolism enzymes in urine of both rats and humans carrying the LRRK2 G2019S mutation. Conversely, these proteins were significantly reduced in inhibitor-treated rats compared to vehicle controls (Figure 4A).

This supports their use as pharmacodynamic markers of LRRK2-targeted therapy in urine. We further clarified in lines 359 – 369:

we trained a machine-learning model on urinary proteins consistently altered by Lrrk2 perturbation in rats (Figure 4C). Applied to human urine, the model effectively distinguished LRRK2 mutation carriers from non-carriers across independent cohorts, indicating conservation of LRRK2-regulated pathways across species and underscoring the translational utility for monitoring LRRK2 pharmacodynamics and supporting ongoing inhibitor trials.

5. *In the intro it is highlighted that LRRK2 may play a role in both genetic and sporadic PD, but the LRRK2 mutation biomarkers identified did not discriminate sporadic PD, only carriers of LRRK2 mutations? LRRK2 is not actually therefore involved in sporadic PD?*

We thank the reviewer for raising this point and the opportunity to clarify an important distinction. The fact that our biomarker model discriminates LRRK2 mutation carriers but not sporadic PD does not imply that the LRRK2 pathway is uninvolved in sporadic disease. Rather, it reflects that our model was trained specifically to detect the proteomic signature of genetic LRRK2 activation (changes to lines 270 – 285).

Importantly, among individuals without known LRRK2 mutations, several exhibited the same proteomic alterations characteristic of G2019S carriers, including seven clinically diagnosed PD or prodromal cases (lines 270–285). These findings are consistent with previous studies indicating that

LRRK2 pathway dysfunction can occur in both genetic and idiopathic PD. Our results therefore suggest that unbiased proteomic profiling may identify subsets of sporadic PD patients with LRRK2-like pathway activation—potentially relevant for targeted therapy even in the absence of mutations. To avoid overinterpretation, we have refined the Introduction and Discussion to delineate these mechanistic and diagnostic aspects, and we now include Appendix Figure S7 illustrating this additional analysis.

Reviewer 2

Major Remarks

1. *In the final section of "Results", the authors suggest that "pathogenic LRRK2-regulated urinary proteins can serve as pharmacodynamic biomarkers". This claim implies that the biomarker candidates identified from human urine cohorts should also exhibit consistent changes in the rat experiments. A key piece of evidence to support this would be a strong correlation between the log2 fold change of significantly changed proteins in LRRKG2019S vs. normal in human cohort and the log2 fold change of mapped rat orthologs in transgenic Lrrk2G2019S vs. WT in rats. However, this correlation is not provided. Simply circling orthologs in rat experiments, as shown in Figure 3C-F, is insufficient, as it only represents protein ID mapping without a quantitative consistency. Therefore, to strengthen the claim, the authors should provide additional analyses demonstrating whether proteomic changes caused by LRRK2 mutation in human cohorts are reproducible in rats.*

We thank the reviewer for highlighting the importance of our quantitative comparison between rat and human data. Effect-size correlation across species and biofluids is expected to be attenuated by measurement noise, cohort heterogeneity (age, sex, co-morbidities, risk variants), matrix differences (urine vs tissue), and incomplete ortholog coverage. To address this rigorously, we estimated adjusted effects in humans using linear models that control for age, sex, and disease status, and compared these standardized coefficients to rat log2 fold-changes.

We now report the requested cross-species effect-size comparison which revealed a modest Pearson R of 0.26, consistent with the known attenuation noted above (Revision Fig. 2A) Importantly, 33 proteins showed elevated levels in both species, including lysosomal proteins GLB1, APOM, FUCA1, ARSB, GALNS, SMPD1, HEXA, CTSA, CTSB, GM2A, SCPEP1, NEU1, DPP7, and CTSZ—demonstrating robust conservation of the signature reflecting lysosomal dysfunction.

We also compared the directionality of regression coefficients (human) with fold-changes (rat), revealing that (changes in lines 355 – 369):

12 out of 30 potential human biomarkers present in rats showed consistent directional regulation—largely upregulated in both G2019S models and downregulated upon LRRK2 inhibition (Figure 4A).

We built a machine learning model using 15 proteins consistently altered by all Lrrk2 modulations in rat urine, then applied it to human PPMI data. The model achieved ROC AUC of 0.75 for distinguishing LRRK2 mutation carriers (Revision Figure 2B; Figure 4C), demonstrating that rat-derived features effectively classify human LRRK2 genetic status.

Revision Figure 2. Comparison of rat and human *LRRK2* G2019S-regulated proteins in urine

A) Correlation of \log_2 fold-changes (G2019S/WT) between rats and humans. Blue data points represent proteins that are significantly regulated (p -value < 5%) in both species. Overlap of significantly elevated proteins in the urine of G2019S carriers in rats and humans, illustrating shared protein regulations are shown in the box.

B) Performance of the predictive model trained on significantly regulated proteins from rats. The model was applied to human urinary proteome data to classify *LRRK2* mutation status.

- Also in the last section of "Result", the analyses of kidney, lung, and brain tissue proteome from rat experiments are minimal. This is a critical limitation, as a more comprehensive mechanistic understanding of PD and *LRRK2* mutations could be derived from examining proteomic changes in these tissues. However, the manuscript lacks in-depth bioinformatic or functional analyses of rat tissue samples to elucidate how *LRRK2* mutations or inhibitor treatments affect the tissue proteomes. Additionally, the authors mention "urinary proteome changes were not fully reflected in tissue proteomes" but provided little discussions to support this statement. Therefore, the authors should conduct a deeper interpretation of their rat tissue data, especially in relation to kidney tissue, since the physiological connection between urine and kidney.

We thank the reviewer for their thoughtful comment. In response, we have expanded the Results and Discussion sections to include a more comprehensive and detailed bioinformatics analysis of the rat tissue proteome (Figure 5, EV4-5). Specifically, we performed extensive statistical analysis to evaluate the effects of *Lrrk2* loss, expression of the G2019S variant, and inhibition of *Lrrk2* kinase activity on the proteomes of the kidney, lung, and brain. Subsequently, we conducted correlation and enrichment analyses to explore tissue-specific and shared impacts of these different *Lrrk2* perturbations. We validated our findings by comparing our results to published work (Kluss et al., 2021). Additionally, we compared these findings with the changes observed in the urinary proteomes of the same rats. Finally, we identified proteins consistently altered by all *Lrrk2* perturbations in rat urine and tested whether their urinary proteomic profiles in urine could classify *LRRK2* mutation status. We have included the above changes in lines 355-369, 381-426 and 527-548.

Minor remarks

1. *The multiple linear regression model for altered protein discovery is not well-described. The authors mention "We employed MLR to account for confounding factors like age and sex affecting protein intensity. (line 110)" along with some details in "Method" section. However, the full set of variables included in the MLR is unclear. The authors should provide a pseudo-equation to explicitly list all variables, for example: Protein log2 intensity ~ age + sex + LRRK2 mutation status + GBA mutation status + PD or not + Additionally, the authors should specify whether different sets of variables were used across different results.*

We have updated the description of the MLR in the methods section according to the reviewer's request (lines 708-715). Specifically, we included the equation used for the MLR and provided a detailed explanation of the sets of variables incorporated into our model:

$$y = \beta_0 + \beta_1 X_1 + \beta_2 X_2 + \beta_3 X_3 + \dots + \beta_n X_n$$

2. *The inference "The smaller number of significant proteins for LRRK2R1441G likely reflects the smaller R1441G cohort (26 carriers) compared to the G2019S cohort (305 carriers) in PPMI. (line 114)" lacks solid support. The authors should firstly consider whether these two mutations have different biological effects. It is possible that LRRK2R1441G exerts a milder influence on LRRK2 function, whereas LRRK2G2019S has a more pronounced effect. The authors provide evidence that these two mutations impact LRRK2 similarly before attributing the difference solely to the sample sizes.*

We appreciate the reviewer bringing this to our attention. Indeed, we cannot solely attribute this difference to the sample sizes. Indeed, G2019S and R1441G mutations could lead to LRRK2 kinase hyperactivation through distinct mechanisms. To address this concern, we mentioned in the revised manuscript that the smaller number of significant proteins in the R1441G cohort may be influenced by both biological differences between the mutations and the smaller cohort size. While both would have to lead to phosphorylation of substrates like Rab GTPases, there is not a clear understanding on the precise mechanisms on how both mutants affect kinase activation. We changed lines 434-441 accordingly:

G2019S, located in the DYG motif of activation loop of the kinase domain, is predicted by metadynamics to directly enhance kinase activity by stabilizing its active conformation (Liu et al., 2013), which would lead to excessive phosphorylation of substrates like Rab GTPases. In contrast, R1441G, in the ROC GTPase domain, destabilizes the ROC-COR interface, potentially affecting LRRK2 tertiary/quaternary structure, and has been reported to show impaired GTPase function, and both of these effects may indirectly promote kinase activation.

3. *Some methodological terms used in the "Results" section are inconsistent with those in the "Method" section, which may cause confusion. For example:*
 - *"Regularized linear regression" in line 160 should explicitly state "Ridge regression" as mentioned in the "Method" section.*

- "K-fold cross validation" in line 205 should match the more detailed description in the "Method" section: "a repeated stratified fold with 5 splits and 3 repeats".

We now incorporate a more detailed description of the linear regression and k-fold cross-validation into the result and method section in lines 219, 228 and 250.

4. In line 217, the authors stated, "The same ML strategy used to differentiate between PD patients and controls, as well as genetic PD and non-manifesting carriers, performed poorly, only slightly better than random chance." However, they do not provide any evidence to support this claim. The authors should include relevant performance metric ROC curves and AUC values as supplementary materials.

We have now included the ROC AUC, for the machine learning analysis to distinguish between PD patients and controls, as well as genetic PD and non-manifesting carriers as Appendix Figure S8A-B (changes in lines 287-289).

5. The authors should provide some information about the mechanisms of action of Lrrk2 inhibitors, which could be helpful to explain the results compared to Lrrk2KO.

We added a brief description on how the Lrrk2 inhibitors attenuate Lrrk2 activity in the results section (lines 329-336).

6. For correlation plots like Figure 3C-F, the author should provide the global correlation for each comparison. For example, in Figure 3C, the Pearson correlation for significantly changed proteins is -0.73. If the correlation of all proteins is near 0, then -0.73 represents a reasonable strong negative correlation. However, if the global correlation is already around -0.6, -0.73 may not be as meaningful in demonstrating a strong negative correlation.

We have included two correlation metrics: "filtered Pearson r" which represents the Pearson correlation of significant proteins in both comparisons (as shown on x and y axes) and "overall Pearson r" which indicates the Pearson correlation of all proteins without prior filtering. We extended the analysis to our tissue dataset. The changes can be seen in Appendix Figure S13.

7. There are less descriptions on MLI-2 results. The authors should consider adding two supplementary correlation figures: Lrrk2KO vs. MLI-2; Lrrk2G2019S vs. MLI-2.

We have now added Lrrk2 KO vs. MLI-2 as well as Lrrk2 G2019S vs. MLI-2 correlations to Appendix Fig. S13 and discussed the results in the discussion section.

8. The caption of Figure EV4 "Peripheral effects of Lrrk2 deficiency, hyperactivation, and inhibition in rat kidney, lung, and brain" is inaccurate, since Figure EV4A and 4B are the plots for urine samples.

We apologize for the confusion. The manuscript has now been substantially revised and the main and EV figures have been restructured for improved clarity. Kidney proteomics data are now presented in Figure 5, while lung and brain proteomics are shown in Figures EV4 and EV5.

9. *The color scheme of kidney, lung, and brain proteins in Figure 3 H-K is confusing. Initially, it appears that many kidney proteins are shown in blue (suggesting down-regulation) despite having a positive log₂ fold-change, or in red (suggesting up-regulation) despite having a negative log₂ fold-change. Upon closer look, it becomes clear they are significantly changed proteins found in urine samples then mapped to other tissues. To improve clarity, the authors should refine the color labels to make them more specific (e.g., "downregulated in urine samples"), or consider alternative visualization methods, such as heatmaps, to better illustrate the data.*

We thank the reviewer for pointing this out. In revising and restructuring the manuscript, we have removed the original figure and replaced it with separate figures dedicated to each biofluid and tissue analysis (Figure 3 for urine, Figure 5 for kidney, and Figure EV4 for lung and Figure EV5 for brain).

10. *Authors should carefully proofread the manuscript to correct minor typographical and formatting errors. Examples include: "LRRK2 inhibitors" in "Summary" should be "Lrrk2 inhibitors"; "2" in "log₂" should be subscripted in the main text and figures.*

We thank the reviewer for identifying the errors in the manuscript; these have been corrected in the revised version.

Reviewer 3

Major remarks:

1. *This study includes multiple cohorts which may encounter variations in sample collection, storage, processing and environmental factors, particularly when the sizes of the three cohorts are quite different. Was there any batch effect correction performed for the proteome datasets? Given that the batch effect is a common factor for clinical proteomics spanning multiple cohorts, detailed elaboration on the batch correction processing and additional discussion on potential batch effects, sample processing variability, and cross-cohort reproducibility would be beneficial. The Authors may need to add the information in Supporting Information.*

We agree that batch-to-batch variations are a prevalent challenge in clinical proteomics. While PPMI sample collection was meticulously controlled and documented (<https://www.ppmi-info.org/access-data-specimens/download-data>), we lacked similarly detailed information for the smaller cohorts, especially LCC, as previously discussed (Virreira Winter et al., 2021). The LCC cohort had younger healthy controls, less rigorous collection protocols, and lacked metadata such as disease severity scores. Additionally, instrumental differences existed: LCC and Columbia used Q Exactive HF-X Orbitrap, while PPMI used the more sensitive timsTOF Pro 2, resulting in disparate protein identifications—over 6,000 proteins in PPMI versus over 2,000 in Columbia and LCC.

To address these differences, our primary analysis investigating pathogenic LRRK2-associated proteome alterations was deliberately conducted separately for each cohort, and the results subsequently compared (Figure 1). This circumvented batch correction necessity and enabled

addressing distinct confounding factors within each cohort, leveraging smaller cohorts as discovery sets validated in the larger, more powerful PPMI cohort.

We integrated the data from all three cohorts only by machine learning to identify discriminating features. Despite batch variations (Appendix Figure S10A), we identified consistent *LRRK2* G2019S-associated signatures robust across independent cohorts, instruments, and protocols (changes to lines 295-307):

We compared feature extraction with and without batch correction using pyComBat (empirical Bayes method). The 30 most discriminating features after batch correction were: SCPEP1, PIK3IP1, NGFR, PFKM, KIFAP3, SLC3A1, QPRT, SDCBP, ENPEP, CCN5, IGKV3D-15, NEU1, COL12A1, CD59, THBD, SGSH, SPP1, DPP7, CTSH, PLD3, RAB12, CTSD, IGKV2-40;IGKV2D-28, CTSB, PLBD2, ATG3, ALPI, PRCP, and SPPL2A. One-third of features overlapped between batch-corrected and non-corrected data. Importantly, lysosomal proteins—NEU1, ALPI, CTSB, ENPEP, and CTSH—remained among the most discriminating factors in both approaches (Appendix Figure S10B). Nearly all batch-corrected features were also identified as differentially abundant in PPMI linear regression, with half represented in the network (Appendix Figure S10B).

Despite PPMI's deeper proteome depth, our analysis showed no bias toward PPMI—almost every model feature was present in LCC and Columbia datasets. The consistent performance across corrected and uncorrected datasets, along with feature overlap and predictive ability, reinforces confidence in our results (Appendix Figure S10C). Methods, results, and Appendix Figure S10 have been added to the revised manuscript accordingly.

- 2. The study successfully identifies LRRK2-associated urinary biomarkers. How would these biomarkers compare to existing PD diagnostic/prognostic markers in cerebrospinal fluid or blood? How do these urinary biomarkers compare with other established LRRK2 activity biomarkers (e.g., phosphorylated Rab proteins in peripheral blood cells)?*

We appreciate the reviewer's recognition that our study has identified pathogenic *LRRK2*-associated urinary biomarkers. To compare our urinary biomarkers with *LRRK2*/PD markers in other biofluids, we have now analyzed the DIA-MS and Olink plasma and CSF datasets from the PPMI cohort (projects 177 and 9000) (Athieniti et al., 2025; Rutledge et al., 2024), as mentioned in response to reviewer 1 comment above. The CSF and plasma Olink (project 9000) consisted of 208 and 227 patient samples respectively, while the DIA-MS CSF and plasma (project 177) included 386 and 178 patient samples respectively. The DIA-MS analyses of plasma and CSF samples had limited proteome depth, with only 186 and 291 proteins identified, respectively (Revision Figure 1A). This stark contrast in proteome depth across biofluids is notable, as only 2.91% and 4.5% of the urinary proteins were detected in plasma and CSF (Revision Figure 1B). Consequently, this difference posed a significant challenge to comparative analyses. Despite these limitations, a comparison between G2019S carriers and non-carriers was conducted, revealing a small proteomic change, where only 4 and 13 proteins in plasma and CSF are significantly regulated (Revision Figure 1C). Proteins of the cathepsin family, specifically CTSB and CTSD, are found to be significantly altered in abundance in *LRRK2*^{G2019S}-dependent manner in both urine and CSF. This aligns with our previous research on CSF proteomics in PD (O. Karayel et al., 2022).

Considering the broader proteome coverage reported, we also analyzed the Olink studies for PPMI. Olink, a proximity extension assay technology, provides high specificity and sensitivity, particularly for plasma analytes. However, it is worth noting that Olink's proteome coverage is more targeted and less "unbiased" than our mass spectrometry-based approach in that many relevant proteins were missing from the panel, thus rather presenting a complementary approach. We have performed the same comparisons using the Olink data and found that urine covers the majority of proteins detected in plasma and CSF, with approximately two-thirds of the Olink panel proteins detectable in urine (Revision Figure 1D). The Olink analysis identified 17 and 56 significantly regulated proteins in the plasma and CSF of LRRK2^{G2019S} carriers, respectively, yet notably, none of the 177 network genes consistently regulated across three urinary cohorts were detected in these studies (Revision Figure 1E). Furthermore, cross-study comparisons with published datasets, including Rutledge et al.'s Olink and SomaScan data (Rutledge et al., 2024), revealed only two proteins (VEGFA and SERPINA11) consistently regulated across multiple studies and biofluids.

Additionally, we would like to emphasize that our study is the first to integrate such a large number of biofluid samples across multiple cohorts to investigate potential protein markers associated with genetic PD. This could hopefully serve as a blueprint for future clinical proteomics investigations.

To address the query regarding comparison with established markers, we analyzed our new targeted urinary proteomics data from the PPMI cohort, specifically measuring levels of LRRK2-phosphorylated RAB10 T73 and other known LRRK2 phosphorylation sites (S910, S935, and S973). After removing outlier samples ($>\pm 3$ SD from median; Appendix Figure 15A) and normalizing to LRRK2 WT median intensity (Appendix Figure 15B). In the revised manuscript we write (lines 510-516):

The results showed elevated levels of both unmodified and phosphorylated peptides in G2019S urine. This aligns with our global urinary proteomics showing increased RAB10 (coefficient 0.48, q-value 2.6e-5) and LRRK2 (coefficient 0.63, q-value 2e-6) protein levels in G2019S carriers (EV3B, Dataset EV1), corroborating previous studies (Fraser et al., 2016; Taymans et al., 2023). While Rab8/10 abundance increased in urinary EVs, phosphorylation rates were unchanged (Taymans, Fell, et al., 2023). These results suggest that, unlike in peripheral blood cells, phosphorylated RAB10 and LRRK2 may not be an effective marker for distinguishing LRRK2 mutation carriers in urine.

Importantly, more than half of our 30 most discriminating proteins showed higher abundance than LRRK2 and RAB10 (Appendix Figure 15C), were considerably easier to quantify, and did not require specialized targeted mass spectrometry.

- 3. Given that urine reflects kidney function and systemic metabolic processes, there are potential confounders and urinary biomarkers may not have high specificity to reflect LRRK2 activity in neurons rather than renal pathology or systemic effects. Were there any correlations between biomarker levels and renal function markers in the dataset? Additionally, since urine contains proteins filtered from multiple tissues, a more explicit discussion on whether these changes specifically reflect neuronal LRRK2 activity versus systemic alterations is needed.*

We appreciate the reviewer's question regarding the specificity of urinary proteomic signatures and whether they reflect neuronal LRRK2 activity versus renal or systemic effects. We agree that

observed protein changes may reflect both direct and indirect consequences of LRRK2 dysfunction.

Recent studies have highlighted urine as a promising, clinically viable matrix for PD biomarker discovery due to non-invasive and repeatable collection (Alcalay et al., 2020; Virreira Winter et al., 2021). However, the extent to which neurodegenerative processes are reflected in urine remains poorly understood. To systematically address this, we now traced the likely tissue origins of all urinary proteins, confirming that the urinary proteome includes proteins from proximal tissues (kidney, bladder) and plasma-derived proteins filtered from distal organs (Revision Figure 3A).

Narrowing this down to proteins altered in pathogenic *LRRK2* carriers (Revision Figure 3B-C), we identified several with brain-specific expression patterns: ABAT, ARSF, ASPDH, BRSK1, CADM2, CADM4, KIAA1549L, MAP4, MARCKSL1, SPOCK3, SPP1, TUBB2A, EPHB1, and JAKMIP3. Additionally, network proteins CHCHD10, CHMP2B, CTSD, FTL, HEXA, HEXB, MME, NAGLU, PLD3, PPT1, PRDX3, RAB10, SMPD1, and TMEM106B are expressed across multiple tissues and implicated in neurodegenerative diseases (Human Protein Atlas, retrieved June 2025), suggesting urinary alterations may indirectly reflect LRRK2-related changes in neuronal tissue.

LRRK2 is highly expressed in blood cells and kidney, and knockout studies implicating it in renal physiology and dysfunction. *Lrrk2* KO mice show autophagy–lysosomal impairment in kidneys with vacuolation (Baptista et al., 2013; Boddu et al., 2015; Kluss et al., 2021; Thévenet et al., 2011; Tong et al., 2010; Zhang et al., 2023). In the revised manuscript we write (lines 541-548):

Accordingly, our revised analysis of the rat urinary proteome showed strongest concordance with kidney; however, a substantial number of urinary proteins were co-regulated in lung (n=137) and brain (n=117) (Appendix Figure S12E). This indicates that urine reflects not only kidney-derived signals but also systemic effects and brain-relevant biology. Moreover, in human urine, pathogenic LRRK2 regulation overlapped with 24 brain proteins that were concomitantly regulated upon Lrrk2 modulation in rat (Figure EV5F), further supporting that urinary proteomics captures brain-relevant consequences of LRRK2 dysfunction.

We also correlated renal function markers (creatinine [CKM], cystatin C [CTS3], albumin [ALB], beta-2 microglobulin [B2M], alpha-1 microglobulin [AMBP], Interleukin 18 [IL-18], Uromodulin [UMOD], Klotho [KL]) with our network and model proteins using Pearson coefficients. LRRK2-associated biomarker changes showed only weak correlation with renal function markers, and correlations were indistinguishable between G2019S and non-G2019S patients (Revision Figure 4A-B), suggesting the observed changes are not representative of renal pathology.

Revision: Figure 3. Annotating urinary proteins using the Human Protein Atlas database.

A. Percentage distribution of urinary proteins based on their tissue-specific expressions (left) and disease associations (right) in the PPMI dataset.

B. Network features (total 177), presented similarly to (A).

C. Machine learning model features (total 30), presented similarly to (A).

Revision Figure 4. Correlating $LRRK2^{G2019S}$ -regulated proteins with kidney markers

A-B) Distribution of Pearson correlation coefficients of 30 machine learning model features (A) and 177 network genes (B) with each kidney marker.

Minor remarks:

1. *For this large-scale study, quality controls for sample collection and data generation in LC-MS/MS workflow are critical. Adding information on quality control measures used will be helpful to ensure reproducibility.*

We employed stringent quality control procedures for both sample processing and LC-MS DIA measurements across all datasets. Our semi-automated sample preparation workflow across all three studies, enabled reproducible quantification of thousands of proteins from minimal urine volumes. Each study utilized pooled control samples as references to evaluate consistency and monitor reproducibility across runs and plates. For LCC and Columbia as detailed in our prior publication (Virreira Winter et al., 2021), repeated measurements showed >90% of proteins with intra- and inter-plate coefficients of variation (CVs) below 50%, and approximately 60% with CVs below 20%. Intra- and inter-plate variability was lower than inter-individual variability, with no protein showing CV below 20%, confirming that proteomic quantification precision significantly exceeded biological variability. We have now expanded the result section to describe similar measures applied to PPMI (lines 140-145):

Pooled references were systematically included as every 12th measured sample (positions A12 to H12 on each 96-well plate). Pooled controls and samples from healthy and diseased individuals were randomized across plates and runs to minimize systemic technical biases. CVs showed high reproducibility with median CV of 68% for patient samples and 25% for pooled samples across plates (Appendix Fig. S2A-B). Principal component analysis showed pooled samples clustered together (Appendix Fig. S2C), and UMAP analysis confirmed minimal instrumental drift and no clustering based on run order (Appendix Figure S1B-C), indicating technical variation does not systematically bias results.

Sample quality filtering: We now implemented biofluid-specific quality marker panels (Geyer et al., 2019) to detect erythrocyte and cellular debris contamination (lines 149-154).

Samples with high contamination markers were excluded. Sample quality was assessed using robust z-scores (Appendix Figure S1F-G). Samples with robust z-scores >3 standard deviations from median

or >1.5 times the interquartile range above the 75th percentile were identified as outliers. These stringent criteria led to removal of 7 samples from Columbia, 4 from LCC, and 43 from PPMI before statistical analysis. Unsupervised analysis revealed sex as the predominant variable driving variance in urine proteomes (Appendix Fig. S1H).

2. Figure 1 (Volcano plots) could benefit from highlighting key proteins rather than showing all 177 proteins.

We are sorry for the confusion. We have actually highlighted the key proteins in the Figure 1 volcano plot. We have clarified this now in the figure legend.

3. Figure 3 (panel C-F). The X- and Y-axis titles can be further revised for better clarity.

We thank the reviewer for pointing this out. We have now revised the titles for better clarity.

References

- Alcalay, R. N., Hsieh, F., Tengstrand, E., Padmanabhan, S., Baptista, M., Kehoe, C., Narayan, S., Boehme, A. K., & Merchant, K. (2020). Higher Urine bis(Monoacylglycerol)Phosphate Levels in LRRK2 G2019S Mutation Carriers: Implications for Therapeutic Development. *Movement Disorders: Official Journal of the Movement Disorder Society*, 35(1), 134–141. <https://doi.org/10.1002/mds.27818>
- Athieniti, E., Afxenti, S., Minadakis, G., & Spyrou, G. M. (2025). *Multi-omics dissection of Parkinson's patients in subgroups associated with motor and cognitive severity* (p. 2025.02.20.638984). bioRxiv. <https://doi.org/10.1101/2025.02.20.638984>
- Baptista, M. A. S., Dave, K. D., Frasier, M. A., Sherer, T. B., Greeley, M., Beck, M. J., Varsho, J. S., Parker, G. A., Moore, C., Churchill, M. J., Meshul, C. K., & Fiske, B. K. (2013). Loss of Leucine-Rich Repeat Kinase 2 (LRRK2) in Rats Leads to Progressive Abnormal Phenotypes in Peripheral Organs. *PLOS ONE*, 8(11), e80705. <https://doi.org/10.1371/journal.pone.0080705>
- Boddu, R., Hull, T. D., Bolisetty, S., Hu, X., Moehle, M. S., Daher, J. P. L., Kamal, A. I., Joseph, R., George, J. F., Agarwal, A., Curtis, L. M., & West, A. B. (2015). Leucine-rich repeat kinase 2 deficiency is protective in rhabdomyolysis-induced kidney injury. *Human Molecular Genetics*, 24(14), 4078–4093. <https://doi.org/10.1093/hmg/ddv147>
- Fraser, K. B., Moehle, M. S., Alcalay, R. N., West, A. B., & LRRK2 Cohort Consortium. (2016). Urinary LRRK2 phosphorylation predicts parkinsonian phenotypes in G2019S LRRK2 carriers. *Neurology*, 86(11), 994–999. <https://doi.org/10.1212/WNL.0000000000002436>
- Geyer, P. E., Voytik, E., Treit, P. V., Doll, S., Kleinhempel, A., Niu, L., Müller, J. B., Buchholtz, M., Bader, J. M., Teupser, D., Holdt, L. M., & Mann, M. (2019). Plasma Proteome Profiling to detect and avoid sample-related biases in biomarker studies. *EMBO Molecular Medicine*, 11(11), e10427. <https://doi.org/10.15252/emmm.201910427>
- Gomes, S., Garrido, A., Tonelli, F., Obiang, D., Tolosa, E., Martí, M. J., Ruiz-Martínez, J., Vinagre-Aragón, A., Hernandez-Eguiaz, H., Croitoru, I., Marshall, V. L., Koenig, T., Hotzy, C., Hsieh, F., Sakalosh, M., Tengstrand, E., Padmanabhan, S., Merchant, K., Bruecke, C., ... Sammler, E. (2023). Elevated urine BMP phospholipids in LRRK2 and VPS35 mutation carriers with and without Parkinson's disease. *NPJ Parkinson's Disease*, 9(1), 52. <https://doi.org/10.1038/s41531-023-00482-4>

- Hadisurya, M., Li, L., Kuwarananchaoen, K., Wu, X., Lee, Z.-C., Alcalay, R. N., Padmanabhan, S., Tao, W. A., & Iliuk, A. (2023). Quantitative proteomics and phosphoproteomics of urinary extracellular vesicles define putative diagnostic biosignatures for Parkinson's disease. *Communications Medicine*, *3*(1), 1–19. <https://doi.org/10.1038/s43856-023-00294-w>
- Heaton, G. R., Li, X., Li, X., Zhou, X., Zhang, Y., Vu, D. T., Oeller, M., Karayel, O., Hoang, Q. Q., Kars, M. E., Kamath, N., Wang, M., Tarassishin, L., Mann, M., Peter, I., & Yue, Z. (2025). Targeting specific kinase substrates rescues increased colitis severity induced by the Crohn's disease-linked LRRK2-N2081D variant. *The Journal of Clinical Investigation*, *135*(19). <https://doi.org/10.1172/JCI190017>
- Hui, K. Y., Fernandez-Hernandez, H., Hu, J., Schaffner, A., Pankratz, N., Hsu, N.-Y., Chuang, L.-S., Carmi, S., Villaverde, N., Li, X., Rivas, M., Levine, A. P., Bao, X., Labrias, P. R., Haritunians, T., Ruane, D., Gettler, K., Chen, E., Li, D., ... Peter, I. (2018). Functional variants in the LRRK2 gene confer shared effects on risk for Crohn's disease and Parkinson's disease. *Science Translational Medicine*, *10*(423), eaa17795. <https://doi.org/10.1126/scitranslmed.aai7795>
- Jennings, D., Huntwork-Rodriguez, S., Henry, A. G., Sasaki, J. C., Meisner, R., Diaz, D., Solanoy, H., Wang, X., Negrou, E., Bondar, V. V., Ghosh, R., Maloney, M. T., Propson, N. E., Zhu, Y., Maciuga, R. D., Harris, L., Kay, A., LeWitt, P., King, T. A., ... Troyer, M. D. (2022). Preclinical and clinical evaluation of the LRRK2 inhibitor DNL201 for Parkinson's disease. *Science Translational Medicine*, *14*(648), eabj2658. <https://doi.org/10.1126/scitranslmed.abj2658>
- Karayel, Ö., Tonelli, F., Winter, S. V., Geyer, P. E., Fan, Y., Sammler, E. M., Alessi, D. R., Steger, M., & Mann, M. (2020). Accurate MS-based Rab10 Phosphorylation Stoichiometry Determination as Readout for LRRK2 Activity in Parkinson's Disease. *Molecular & Cellular Proteomics*, *19*(9), 1546–1560. <https://doi.org/10.1074/mcp.RA120.002055>
- Karayel, O., Virreira Winter, S., Padmanabhan, S., Kuras, Y. I., Vu, D. T., Tuncali, I., Merchant, K., Wills, A.-M., Scherzer, C. R., & Mann, M. (2022). Proteome profiling of cerebrospinal fluid reveals biomarker candidates for Parkinson's disease. *Cell Reports Medicine*, *3*(6), 100661. <https://doi.org/10.1016/j.xcrm.2022.100661>
- Kluss, J. H., Mazza, M. C., Li, Y., Manzoni, C., Lewis, P. A., Cookson, M. R., & Mamais, A. (2021). Preclinical modeling of chronic inhibition of the Parkinson's disease associated kinase LRRK2 reveals altered function of the endolysosomal system in vivo. *Molecular Neurodegeneration*, *16*(1), 17. <https://doi.org/10.1186/s13024-021-00441-8>
- Liu, M., Bender, S. A., Cuny, G. D., Sherman, W., Glicksman, M., & Ray, S. S. (2013). Type II kinase inhibitors show an unexpected inhibition mode against Parkinson's disease-linked LRRK2 mutant G2019S. *Biochemistry*, *52*(10), 1725–1736. <https://doi.org/10.1021/bi3012077>
- Rideout, H. J., Chartier-Harlin, M.-C., Fell, M. J., Hirst, W. D., Huntwork-Rodriguez, S., Leyns, C. E. G., Mabrouk, O. S., & Taymans, J.-M. (2020). The Current State-of-the Art of LRRK2-Based Biomarker Assay Development in Parkinson's Disease. *Frontiers in Neuroscience*, *14*. <https://doi.org/10.3389/fnins.2020.00865>
- Rutledge, J., Lehallier, B., Zarifkar, P., Losada, P. M., Shahid-Besanti, M., Western, D., Gorijala, P., Ryman, S., Yutsis, M., Deutsch, G. K., Mormino, E., Trelle, A., Wagner, A. D., Kerchner, G. A., Tian, L., Cruchaga, C., Henderson, V. W., Montine, T. J., Borghammer, P., ... Poston, K. L. (2024). Comprehensive proteomics of CSF, plasma, and urine identify DDC and other biomarkers of early Parkinson's disease. *Acta Neuropathologica*, *147*(1), 52.

<https://doi.org/10.1007/s00401-024-02706-0>

- Taymans, J.-M., Fell, M., Greenamyre, T., Hirst, W. D., Mamais, A., Padmanabhan, S., Peter, I., Rideout, H., & Thaler, A. (2023). Perspective on the current state of the LRRK2 field. *Npj Parkinson's Disease*, 9(1), 1–9. <https://doi.org/10.1038/s41531-023-00544-7>
- Taymans, J.-M., Mutez, E., Sibrán, W., Vandewynckel, L., Deldycke, C., Bleuse, S., Marchand, A., Sarchione, A., Leghay, C., Kreisler, A., Simonin, C., Koprach, J., Baille, G., Defebvre, L., Dujardin, K., Destée, A., & Chartier-Harlin, M.-C. (2023). Alterations in the LRRK2-Rab pathway in urinary extracellular vesicles as Parkinson's disease and pharmacodynamic biomarkers. *Npj Parkinson's Disease*, 9(1), 21. <https://doi.org/10.1038/s41531-023-00445-9>
- Thévenet, J., Pescini Gobert, R., Hooft van Huijsduijnen, R., Wiessner, C., & Sagot, Y. J. (2011). Regulation of LRRK2 expression points to a functional role in human monocyte maturation. *PLoS One*, 6(6), e21519. <https://doi.org/10.1371/journal.pone.0021519>
- Tong, Y., Yamaguchi, H., Giaime, E., Boyle, S., Kopan, R., Kelleher, R. J., & Shen, J. (2010). Loss of leucine-rich repeat kinase 2 causes impairment of protein degradation pathways, accumulation of alpha-synuclein, and apoptotic cell death in aged mice. *Proceedings of the National Academy of Sciences of the United States of America*, 107(21), 9879–9884. <https://doi.org/10.1073/pnas.1004676107>
- Virreira Winter, S., Karayel, O., Strauss, M. T., Padmanabhan, S., Surface, M., Merchant, K., Alcalay, R. N., & Mann, M. (2021). Urinary proteome profiling for stratifying patients with familial Parkinson's disease. *EMBO Molecular Medicine*, 13(3), e13257. <https://doi.org/10.15252/emmm.202013257>
- Zhang, S., Qian, S., Liu, H., Xu, D., Xia, W., Duan, H., Wang, C., Yu, S., Chen, Y., Ji, P., Wang, S., Cui, X., Wang, Y., & Shen, H. (2023). LRRK2 aggravates kidney injury through promoting MFN2 degradation and abnormal mitochondrial integrity. *Redox Biology*, 66, 102860. <https://doi.org/10.1016/j.redox.2023.102860>

2nd Dec 2025

Manuscript Number: MSB-2025-12976R

Title: Multi-cohort, cross-species urinary proteomics reveals LRRK2 dysfunction in Parkinson's disease

Author: Duc Tung Vu

William Sibran

Andreas Metousis

Laurine Vandewynckel

Basak Eraslan

Liesel Goveas

Ericka Itang

Claire Deldycke

Adriana Figueroa-Garcia

Réginald Lefebvre

Johannes Müller-Reif

Sebastian Virreira-Winter

Marie-Christine Chartier-Harlin

Jean-Marc Taymans

Matthias Mann

Ozge Karayel

Dear Dr Karayel,

Thank you for sending us your revised manuscript. We have now heard back from the reviewer who agreed to evaluate your updated study. As you will see below, the reviewer thinks that the revision has addressed their comments as well as those of the other reviewers.

Before we can proceed with formal acceptance, we kindly ask you to address the following remaining issue:

1. The reviewer's remaining relatively minor comments.

On a more editorial level:

2. Please remove all figures from the manuscript file and upload them as individual figure files. The legends for both the main figures and the EV figures should remain at the end of the manuscript, after the References section.

3. Please remove the "Authors' Contributions" section from the manuscript.

4. The Appendix needs to be uploaded in PDF format.

5. Please note that the funding information is currently missing in the system and needs to be completed. The following funders must be entered separately:

- The Michael J. Fox Foundation (MJFF-12938.4 and MJFF-004693)
- The Max Planck Society for the Advancement of Science
- Scholarships from the Université de Lille (Ecole Doctorale Biologie-Santé)

6. In the "Disclosure Statement and Competing Interests" section:

- All employment in biotech companies should be disclosed.
- Please add the following sentence: "MM is an editorial advisory board member. This has no bearing on the editorial consideration of this article for publication."

7. References:

- Use "et al." after listing 10 author names.
- DOIs should only be included for preprints and datasets that have not yet been published.

8. EV datasets should not be uploaded as zipped files. Instead, please upload each Excel file individually (Dataset EV1 through Dataset EV6) using the file format "Data set."

7. Data availability:

- Please remove the reviewer username and passcode, and ensure that the datasets are made publicly available upon

acceptance of the manuscript. A specific URL for the dataset (ProteomeXchange Consortium PXD057308) should be provided in the data availability statement.

- Was the PPMI data generated in this study? According to the journal's data policy, if feasible and consistent with the individual consent agreement, authors must deposit human clinical datasets in public databases at the time of publication. A corresponding data availability statement should be included in this section.

8. Please address the following issues in figure legends:

- Please note that the box plots need to be defined in terms of minima, maxima, centre, bounds of box and whiskers, and percentile in the legends of figures 1g; 4a

- Please note that information related to n is missing in the legends of figures 1d, e; 3a-d; 5a-d; EV 1e; EV 4a-d; EV 5a-d

9. I have modified the synopsis text (see attached). Please let us know if you are fine with it or if you would like to introduce further modifications.

10. The manuscript sections should be in the following order: Title page - Abstract & Keywords - Introduction - Results - Discussion - Methods - Data Availability - Acknowledgments - Disclosure Statement & Competing Interests - References - Figure Legends - (Main Tables with legends if applicable) - Expanded View Figure Legends.

Please resubmit your revised manuscript online, with a covering letter listing amendments and responses to each point raised by the reviewer.

Kind regards,
Jingyi

Jingyi Hou, PhD
Senior Editor
Molecular Systems Biology

*** PLEASE NOTE *** As part of the EMBO Press transparent editorial process initiative (see our Editorial at <https://dx.doi.org/10.1038/msb.2010.72> , Molecular Systems Biology will publish online a Review Process File to accompany accepted manuscripts. When preparing your letter of response, please be aware that in the event of acceptance, your cover letter/point-by-point document will be included as part of this File, which will be available to the scientific community. More information about this initiative is available in our Instructions to Authors. If you have any questions about this initiative, please contact the editorial office (msb@embo.org).

Reviewer #1:

The authors have provided valuable and comprehensive responses to all reviewers' comments and have made meaningful revisions to the manuscript. As the previous Reviewer #2, I believe the authors have addressed my questions and those of the other reviewers well. I recommend that the manuscript be considered for publication in Molecular Systems Biology after minor revisions. My detailed comments are as follows:

0. One issue I must point out is that the line numbers provided in the authors' response document do not match the line numbers in the revised manuscript. This discrepancy makes it difficult to locate the referenced changes. The authors should be more careful with line-numbering in future revisions.

1. At the editor's request, I have carefully reviewed authors' responses to Reviewer #1's points #2 and #3. For Point #2, the authors clearly described the existing LRRK2-pathway biomarkers and urinary BMPs, and appropriately discussed their limitations. They also indicated the diagnostic value of urine proteomics, underscoring the significance of this study. For Point #3, the authors made reasonable comparisons between their work and previously published studies, effectively highlighting the novelty and advancements of this study. These comparisons also revealed new insights not captured in earlier

CSF or plasma studies. However, the authors did not adequately address Reviewer #1's question regarding how this study supports ongoing LRRK2 inhibitor trials. Additional clarification on this point is needed.

In addition, I identified an error in Revision Figure 1D: the numbers shown in the two Venn diagrams are the same, which appears incorrect. The authors should revise this figure.

2. Regarding my previous major question #1, the Pearson correlation of 0.26 between human and rat urine is low, but author's explanation and further analyses are reasonable. My remaining question is what the total number of consistently altered proteins (i.e., both upregulated and both downregulated) is, and why the authors selected only 15 of these proteins for classification rather than using all consistently altered proteins.

3. About minor point #1, in line 728 of revised manuscript, authors described the linear model: "Where y is the protein intensity, β_0 the intercept, β_1 the coefficient of age, β_2 the coefficient for sex and β_{3-n} is the coefficient for the variable e.g. LRRK2 mutation. X_1-3 are the metadata values including age, sex and mutation status." I am still unclear about the number of variables in this model. If there are only three variables (age, sex, mutation status), then " β_{3-n} " should simply be " β_3 ," consistent with " X_1-3 ."

4. The authors should carefully proofread the revised manuscript again for minor errors. For example, in line 910, the "L" in "log2" should not be capitalized.

January 7, 2026

Response to Reviewers' Comments: Multi-Cohort, Cross-Species Urinary Proteomics Reveals Signatures of LRRK2 Dysfunction in Parkinson's Disease

We thank the reviewer for their re-evaluation of our manuscript. Our point-by-point responses to the remaining suggestions are provided below.

One issue I must point out is that the line numbers provided in the authors' response document do not match the line numbers in the revised manuscript. This discrepancy makes it difficult to locate the referenced changes. The authors should be more careful with line-numbering in future revisions.

We apologize for the inconsistency in line numbering. We have now highlighted all changes mentioned in this response in red in the revised manuscript and included the corresponding page numbers.

At the editor's request, I have carefully reviewed authors' responses to Reviewer #1's points #2 and #3. For Point #2, the authors clearly described the existing LRRK2-pathway biomarkers and urinary BMPs and appropriately discussed their limitations. They also indicated the diagnostic value of urine proteomics, underscoring the significance of this study. For Point #3, the authors made reasonable comparisons between their work and previously published studies, effectively highlighting the novelty and advancements of this study. These comparisons also revealed new insights not captured in earlier CSF or plasma studies. However, the authors did not adequately address Reviewer #1's question regarding how this study supports ongoing LRRK2 inhibitor trials. Additional clarification on this point is needed.

We thank the reviewer for the opportunity to clarify the clinical utility of our findings. This study supports ongoing and future LRRK2 inhibitor trials in three specific ways:

1. Preclinical validation of translatable LRRK2 pharmacodynamic markers: Our study establishes the rat urinary proteome as a high-fidelity platform for the identification and causal validation of LRRK2-linked biomarkers. While current readouts such as pRab10 confirm target engagement, they do not necessarily report on whether the intervention restores the downstream biological consequences of LRRK2 dysfunction. By integrating genetic (gain- and loss-of-function) rat models with pharmacological inhibition (MLi-2 and PF-475), we provide a causal framework demonstrating that LRRK2 activity shapes the urinary proteome.

Specifically, we show that the human LRRK2-associated urinary protein network contains a core signature—including key lysosomal hydrolases such as cathepsins, GM2A, PLD3, HEXB, and NEU1—that is largely recapitulated in rats and, crucially, reversed following kinase inhibition (Figures 3 and 4). This cross-species conservation nominates these proteins as high-confidence, pathway-centric pharmacodynamic markers. In a clinical trial setting,

monitoring this multi-protein signature allows for the assessment of 'pathway normalization'—the successful rescue of systems-level defects in lysosomal function, glycosphingolipid metabolism, and membrane trafficking that characterize LRRK2-driven pathology. Furthermore, these protein-level readouts are likely more resilient to the technical variation and biological noise often encountered with phosphorylation-based markers in biofluids. By providing a stable, scalable and non-invasive readout of biological rescue, our findings support the tiered biomarker strategy proposed in our manuscript: utilizing pRab10/BMPs to confirm target engagement and our urinary protein panel to monitor longitudinal pathway restoration and therapy response.

2. An opportunity for biology-based patient stratification: Current trials often rely solely on genotype (e.g., G2019S carriers). However, our ML model identified "G2019S-like" proteomic signatures in idiopathic PD patients and carriers of other risk variants. This suggests that our urinary protein panel could be utilized as a screening tool to identify "high-LRRK2-activity" patients, regardless of their family history or mutation status. This biology-based stratification implies that LRRK2-targeted therapies could benefit a broader population than just those with familial mutations, thereby expanding the potential trial population and enabling more precise recruitment.

3. Non-invasive monitoring of CNS and systemic effects: A primary challenge in trials for neurodegenerative disorders is the difficulty of frequent, longitudinal CSF sampling. We identified 24 proteins in human urine that are directly regulated in the rat brain following LRRK2 perturbations (Fig. 5F). This establishes urine as a "proxy" matrix that can capture brain-relevant molecular changes non-invasively. This capability could support high-frequency longitudinal monitoring in the late-stage trials to assess dose-response relationships and long-term safety without the burden of repeated lumbar punctures.

We have updated the Discussion section to explicitly emphasize these points (highlighted in red).

In addition, I identified an error in Revision Figure 1D: the numbers shown in the two Venn diagrams are the same, which appears incorrect. The authors should revise this figure.

We appreciate the reviewer's attention to detail regarding Revision Figure 1D and apologize for the confusion. The identical values in the Venn diagrams are correct and a direct result of the study design. Both the CSF and plasma samples were analyzed using the Olink Explore 1463 Panel, which consists of a fixed set of 1,463 protein assays. In this figure, we compared the coverage of this fixed Olink panel against the same discovery-based urine dataset. Because the Olink panel content is identical regardless of the biofluid it is applied to, its overlap with the urine dataset remains constant in both comparisons. This figure was intended to illustrate the coverage of the fixed panel relative to our discovery data, rather than fluid-specific protein detection. We hope this clarifies the matter.

Regarding my previous major question #1, the Pearson correlation of 0.26 between human and rat urine is low, but author's explanation and further analyses are reasonable. My

remaining question is what the total number of consistently altered proteins (i.e., both upregulated and both downregulated) is, and why the authors selected only 15 of these proteins for classification rather than using all consistently altered proteins.

We appreciate the opportunity to clarify our feature selection process. While 41 proteins were significantly altered in both species, we intentionally restricted our machine learning features to the 15 proteins identified at the intersection of all four rat perturbations (mutation, knockout, and two distinct inhibitors). We chose this "rat-to-human" discovery pipeline primarily to avoid data leakage. Selecting features based on their significance in the human datasets prior to testing would introduce circularity and artificially inflate model performance. By restricting feature selection to this high confidence core rat signature, we aimed an unbiased evaluation of the model's true predictive capacity across species. Furthermore, this 15-protein consensus signature ensures that the model captures a robust biological signal consistently linked to LRRK2 pathway activity across both genetic (gain- and loss-of-function) and pharmacological modulations. We have clarified this in the manuscript (Page 11) to provide rationale for feature selection: *"Next, we identified a high-confidence signature of 15 urinary proteins that were consistently regulated across all LRRK2 perturbations in rats (Figure 4B). We used this consensus signature to train a support vector machine on the combined PPMI–LCC–Columbia human urinary proteomics dataset (Dataset EV4). Feature selection was deliberately restricted to these rat-derived features to prevent data leakage and ensure an unbiased evaluation of the model's true predictive capacity across species."*

About minor point #1, in line 728 of revised manuscript, authors described the linear model: "Where y is the protein intensity, β_0 the intercept, β_1 the coefficient of age, β_2 the coefficient for sex and β_{3-n} is the coefficient for the variable e.g. LRRK2 mutation. X_1-3 are the metadata values including age, sex and mutation status." I am still unclear about the number of variables in this model. If there are only three variables (age, sex, mutation status), then " β_{3-n} " should simply be " β_3 ," consistent with " X_{1-3} ."

We appreciate the reviewer's request for clarification regarding the model variables. The notation was intended to reflect that the number of variables (n) changes depending on the specific analysis (e.g., a single LRRK2 mutation versus the 58 PD risk variants). To clarify, for the WGS-derived analysis, the model includes age (X_1), sex (X_2) and the 58 PD risk variants (X_3 through X_{60}) totaling 60 variables. We have revised the manuscript (Page 22) to use more precise notation that accurately reflects the variable count. The definition now reads: *"Where y is the protein intensity, β_0 the intercept, β_1 and β_2 the coefficients for age (X_1) and sex (X_2) and $\beta_{3..n}$ are the coefficients for the $n-2$ remaining variables ($X_{3..n}$), such as the 58 risk variants or specific mutation indicators."*

The authors should carefully proofread the revised manuscript again for minor errors. For example, in line 910, the "L" in "log2" should not be capitalized.

We thank the reviewer for pointing these out. We have carefully proofread the manuscript and corrected the minor errors identified, including the capitalization of 'log₂' for consistency throughout the text.

13th Jan 2026

Manuscript number: MSB-2025-12976RR

Title: Multi-cohort, cross-species urinary proteomics reveals LRRK2 dysfunction in Parkinson's disease

Dear Dr Karayel,

Thank you again for sending us your revised manuscript. We are now satisfied with the modifications made and I am pleased to inform you that your paper has been accepted for publication.

You may qualify for financial assistance for your publication charges - either via a Springer Nature fully open access agreement or an EMBO initiative. Check your eligibility: <https://link.springer.com/journal/44320/how-to-publish-with-us>

Sincerely,
Jingyi

Jingyi Hou, PhD
Senior Editor
Molecular Systems Biology

>>> Please note that it is Molecular Systems Biology policy for the transcript of the editorial process (containing referee reports and your response letter) to be published as an online supplement to each paper. If you do NOT want this, you will need to inform the Editorial Office via email immediately. More information is available here: <https://link.springer.com/partners/embo-press/editorial-policies#Peer%20review>